# Hummingbird: High Fidelity Image Generation via Multimodal Context Alignment

**Minh-Quan Le**[1,2,⋆], **Gaurav Mittal**[1⋆], **Tianjian Meng**[1], **A S M Iftekhar**[1],
**Vishwas Suryanarayanan**[1], **Barun Patra**[1], **Dimitris Samaras**[2], **Mei Chen**[1]
[1]Microsoft, [2]Stony Brook University

## Abstract

While diffusion models are powerful in generating high-quality, diverse synthetic data for object-centric tasks, existing methods struggle with scene-aware tasks such as Visual Question Answering (VQA) and Human-Object Interaction (HOI) Reasoning, where it is critical to preserve scene attributes in generated images consistent with a multimodal context, i.e. a reference image with accompanying text guidance query. To address this, we introduce **Hummingbird**, the first diffusion-based image generator which, given a multimodal context, generates highly diverse images w.r.t. the reference image while ensuring high fidelity by accurately preserving scene attributes, such as object interactions and spatial relationships from the text guidance. Hummingbird employs a novel Multimodal Context Evaluator that simultaneously optimizes our formulated Global Semantic and Fine-grained Consistency Rewards to ensure generated images preserve the scene attributes of reference images in relation to the text guidance while maintaining diversity. As the first model to address the task of maintaining both diversity and fidelity given a multimodal context, we introduce a new benchmark formulation incorporating MME Perception and Bongard HOI datasets. Benchmark experiments show Hummingbird outperforms all existing methods by achieving superior fidelity while maintaining diversity, validating Hummingbird's potential as a robust multimodal context-aligned image generator in complex visual tasks. Project page: `https://roar-ai.github.io/hummingbird`

## 1 Introduction

In recent years, diffusion models (Ho et al., 2020; Rombach et al., 2022) have emerged as powerful tools for image generation, offering impressive capabilities in creating high-quality and diverse synthetic data. This synthetic data has proven valuable in various applications, particularly for *object-centric* image classification tasks (Shu et al., 2022; Feng et al., 2023). However, for *scene-aware* tasks like Visual Question Answering (VQA) (Goyal et al., 2017; Antol et al., 2015) and Human-Object Interaction (HOI) Reasoning (Jiang et al., 2022; Ulutan et al., 2020), it is essential that the generated images accurately preserve scene attributes relevant to the task, as specified by the accompanying text queries. These attributes may include numerical attributes (e.g., object count), physical properties (e.g., color), spatial relationships (e.g., relative positions of objects), object interactions (e.g., how objects interact with each other), or scene-level details. Although defining an all-exhaustive list of such attributes is intractable, tasks like VQA and HOI, through their open-ended questions (Antol et al., 2015), enable to capture the relevant semantic attributes effectively.

While existing image generation methods have made great strides in improving the *diversity* (defined as how different the generated images are from a given reference image and from each other), they often fall short in maintaining high *fidelity*. Given a text query as guidance pertaining to scene attributes in relation to a given reference image (together referred to as *multimodal context*), we define *fidelity* as how truthfully the image generator can preserve those attributes in the generated image. Existing methods, including state-of-the-art (SOTA) diffusion models like Stable Diffusion XL (SDXL) (Podell et al., 2024), Image Translation techniques like Boomerang (Luzi et al., 2024), Textual Inversion (Gal et al., 2023; Trabucco et al., 2024), and Image Variation (Xu et al., 2023),

---
⋆Equal contribution. This work was done as Minh-Quan's internship project at Microsoft.

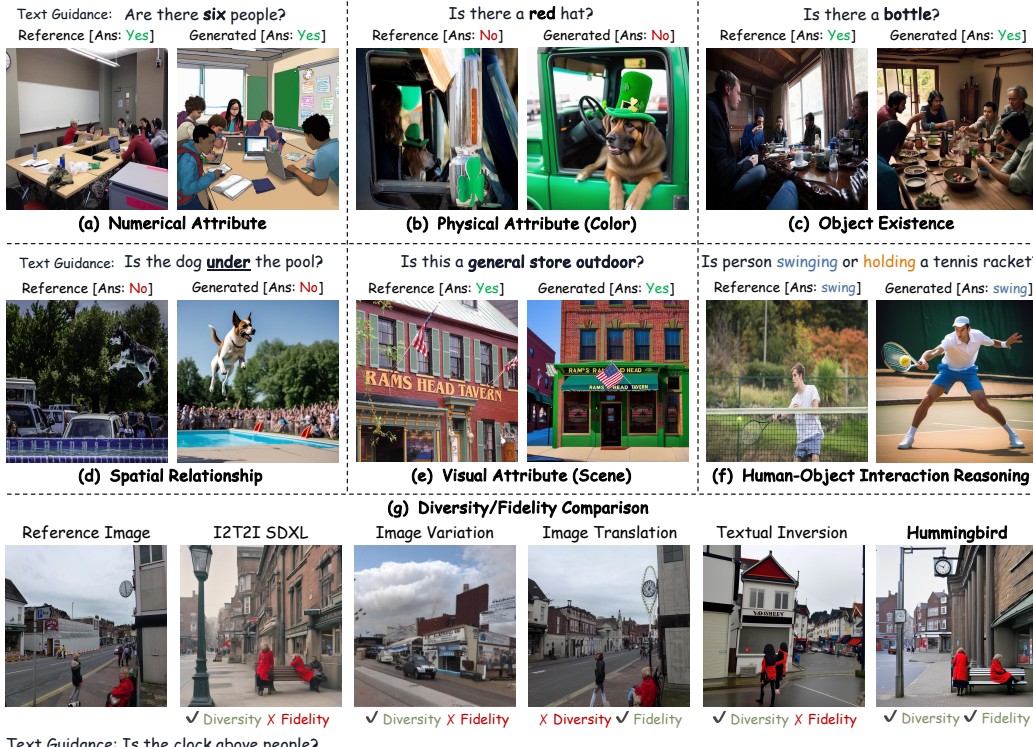

Figure 1: Hummingbird aligns generated image with multimodal context input (reference image + text guidance) ensuring the synthetic image is diverse w.r.t. reference image while exhibiting high fidelity (i.e. preserves the scene attribute from reference image in relation to text guidance). (a-f) By doing so, for VQA and HOI Reasoning, Hummingbird enables the answer to the question in the text guidance to remain consistent between the reference and generated images. (g) Hummingbird is also able to preserve both diversity and fidelity without the trade-off exhibited by existing methods.

often lose fine-grained scene details as the embeddings do not have a precise control on which scene attributes to capture. This lack of fidelity is particularly problematic in synthetic data generation for *scene-aware* tasks like VQA and HOI Reasoning, where preserving scene elements in relation to text query is critical for high performance.

To achieve high fidelity while preserving diversity in image generation w.r.t multimodal context, we introduce Hummingbird, the first diffusion-based general-purpose image generator designed to produce high-fidelity images guided by a multimodal context while maintaining diversity. Given a multimodal context, containing a reference image and accompanying text guidance, Hummingbird generates highly diverse images that differ from the reference image while accurately preserving the scene attributes referenced in the text guidance.

Hummingbird contains a novel Multimodal Context Evaluator that simultaneously maximizes our formulated Global Semantic and Fine-grained Consistency Rewards. Given the multimodal context input, Hummingbird crafts an instruction prompt to provide to a Multimodal Large Language Model (MLLM) (Liu et al., 2024; Chen et al., 2024) to obtain a text-based Context Description containing the necessary scene attributes to focus on for image generation. Hummingbird then finetunes the SDXL diffusion model using the rewards from the Multimodal Context Evaluator based on the alignment between the generated image and the MLLM Context Description. While SDXL enables generating images with high diversity (i.e. visually different from the reference image), the rewards encourage the fine-tuning process to align the generated image closely with the scene attributes provided in the multimodal context to achieve high fidelity. Unlike existing methods, Hummingbird is able to balance both diversity and fidelity in generated images which is a prerequisite for leveraging synthetic data for scene-aware tasks like VQA and HOI Reasoning. Figure 1 (a-f) illustrates how, given text guidance (a question) and a reference image, the generated image remains visually diverse

w.r.t reference image while preserving the specified scene attributes, thus ensuring that the answer to the question in the text guidance remains consistent between the reference and generated images.

As the first approach to address the task of multimodal context-aware image generation with high fidelity and diversity, we also introduce a new benchmark formulation to evaluate different methods on this task. Our benchmark leverages the MME Perception benchmark (Fu et al., 2024) to perform Test-Time Augmentation (TTA) (Shanmugam et al., 2021; Kim et al., 2020) with real and generated synthetic images to evaluate the ability of a method to maintain fidelity on scene attributes related to spatial existence, count, position, color, and scene. We further leverage Bongard Human-Object Interaction (HOI) (Jiang et al., 2022) dataset to perform Test-time Prompt Tuning (TPT) (Shu et al., 2022) to test a method's ability to maintain fidelity when focusing on sophisticated human-object interactions. Finally, we compute a method's ability to maintain diversity using feature-based distance metrics between the reference and generated images. Experiments show that Hummingbird is able to outperform all existing methods on MME Perception and Bongard HOI (i.e. provide the best fidelity) while achieving high diversity. Hummingbird also outperforms all other methods consistently on ImageNet and its OOD variants. This validates its effectiveness on large and diverse datasets from both scene-aware and object-centric tasks. Figure 1 (g) demonstrates that Hummingbird is able to preserve diversity and fidelity without the trade-off exhibited by existing methods.

Our contributions are as follows:

- We introduce Hummingbird, the first diffusion-based image generator to synthesize high fidelity images guided by multimodal context of reference image and text guidance while maintaining high diversity.

- We develop a novel Multimodal Context Evaluator that simultaneously maximizes our formulated Global Semantic and Fine-grained Consistency Rewards during end-to-end fine-tuning. This enables the diffusion model to focus on both local and global scene attributes.

- Hummingbird outperforms existing methods on MME Perception and Bongard HOI datasets as part of a newly formulated benchmark that evaluates the ability to generate images with both high fidelity and high diversity given a multimodal context.

## 2 RELATED WORK

Popular diffusion-based image generators, such as **SDXL** (Podell et al., 2024), have made tremendous progress in producing high-quality diverse images. But these methods struggle to fully translate complex scene details from text to image, especially if it involves preserving a specific scene attribute w.r.t. a reference image. We can observe this limitation in Figure 1(g) where the image generated based on a text description of a reference image (image-to-text-to-image (**I2T2I**)) cannot preserve the scene elements from reference image (poor fidelity).

**Image Variation** methods like Xu et al. (2023); Feng et al. (2023); Zhang et al. (2024); Le et al. (2024); Graikos et al. (2024); Belagali et al. (2024), use an image encoder to extract embeddings to condition diffusion models. While this helps to retain high-level semantics and allow changes to style and other low-level features, these methods fail to preserve specific scene attributes of a reference image due to a lack of precise control over which detail to emphasize. Moreover, addition of noise during diffusion process introduces randomness which causes significant deviation from the reference image, again resulting in poor fidelity (Figure 1 (g)).

**Image Translation** or Image Editing techniques, such as Boomerang (Luzi et al., 2024) and ControlNet (Zhang et al., 2023), add noise to reference input image to create variations around the original reference image in latent space. Since they aim to generate multiple local samples, they fail to produce sufficiently diverse images (Figure 1 (g)). Moreover, the added noise during forward diffusion step often distorts or removes the details on specific scene attributes in the reference image.

**Textual Inversion** methods (Gal et al., 2023; Trabucco et al., 2024; Ahn et al., 2024; Nguyen et al., 2024) encode new concepts as "pseudo-words" in text embedding space to guide image generation. Although effective for style transfer, these methods struggle with tasks where a single reference image is involved, such as TTA, to generate multiple synthetic images. When fine-tuning with just one reference image, these methods lead to images containing irrelevant scene attributes and fail to preserve the attributes from the original reference image (leading to poor fidelity, Figure 1 (g)).

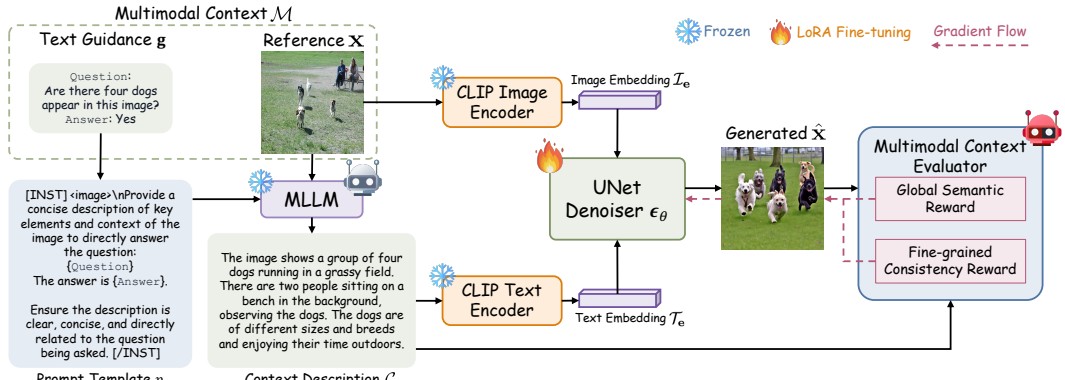

Figure 2: **Model Overview**: Given text guidance **g** and reference image **x** (multimodal context $\mathcal{M}$), Hummingbird crafts an instruction prompt $p$ to feed to MLLM and obtain Context Description $\mathcal{C}$. It them embeds **x** and $\mathcal{C}$ via CLIP to feed to UNet Denoiser of SDXL to generate image $\hat{\mathbf{x}}$. To improve the fidelity of $\hat{\mathbf{x}}$ w.r.t. $\mathcal{M}$ while preserving diversity, Hummingbird introduced Multimodal Context Evaluator to simultaneously maximize novel rewards – Global Semantic and Fine-Grained Consistency Rewards – to align $\hat{\mathbf{x}}$ with scene attributes provided in $\mathcal{M}$.

## 3 PRELIMINARIES

**Latent Diffusion Models (LDM)** (Podell et al., 2024; Rombach et al., 2022) operate in a compressed latent space rather than directly in pixel space. Using a VAE (Kingma, 2013) encoder $\mathcal{E}$, they encode an input image $\mathbf{x} \sim \mathbb{P}_{\text{data}}$ into a lower-dimensional latent variable $\mathbf{z}_0$, on which the diffusion process is applied. The model iteratively corrupts $\mathbf{z}_0$ with noise and learns to reverse this process to generate new samples coherent with the original data distribution. The training objective minimizes the error between the noisy latent sample at time $t$ and the denoised prediction:

$$\mathcal{L}_{\text{simple}} = \mathbb{E}_{t \sim \mathcal{U}[0,1], \mathbf{x} \sim \mathbb{P}_{\text{data}}, \mathbf{z}_0 = \mathcal{E}(\mathbf{x}), \epsilon \sim \mathcal{N}(\mathbf{0}, \mathbf{I})} \left[ \| \epsilon - \epsilon_\theta(\mathbf{z}_t, t) \|^2 \right], \tag{1}$$

where $\epsilon_\theta(\mathbf{z}_t, t)$ represents UNet denoiser's (Ho et al., 2020; Rombach et al., 2022) predicted noise.

**Denoising Diffusion Implicit Models (DDIM)** (Song et al., 2021) accelerate the sampling process in diffusion models while maintaining high-quality samples. DDIM adjusts the reverse diffusion process by directly predicting the latent variable at the next timestep using deterministic sampling, which allows for faster convergence. The DDIM update step is defined as:

$$\mathbf{z}_{t-1} = \sqrt{\alpha_{t-1}} \frac{\mathbf{z}_t - \sqrt{1 - \alpha_t} \epsilon_\theta(\mathbf{z}_t, t)}{\sqrt{\alpha_t}} + \sqrt{1 - \alpha_{t-1}} \cdot \epsilon_\theta(\mathbf{z}_t, t). \tag{2}$$

## 4 METHOD

### 4.1 MODEL OVERVIEW

Figure 2 provides an overview of the end-to-end training setup of Hummingbird. Let $\mathcal{M} = \{\mathbf{x}, \mathbf{g}\}$ denote the multimodal context fed as input to Hummingbird. $\mathcal{M}$ comprises of the reference image $\mathbf{x}$ and text guidance $\mathbf{g}$ for Hummingbird to generate image $\hat{\mathbf{x}}$ that is visually diverse w.r.t. $\mathbf{x}$ while faithfully preserves the scene attributes in relation to $\mathbf{g}$. Depending on the task, $\mathbf{g}$ can be a question or a set of annotations accompanying $\mathbf{x}$. During training, $\mathbf{g}$ additionally consists of the ground truth (such as answer to the question or correct annotation) which is unavailable during evaluation.

We feed the multimodal context $\mathcal{M}$ to MLLM to generate a text-based Context Description, $\mathcal{C}$. For this, we need to provide $\mathcal{M}$ as an instruction prompt to MLLM such that it guides the MLLM on which scene attributes to focus on in the reference image $\mathbf{x}$ in relation to the text guidance $\mathbf{g}$. We devise a prompt template $p : \mathbf{g} \rightarrow \mathcal{P}$ to transform $\mathbf{g} \in \mathcal{M}$ into an instruction prompt $\mathcal{P}$. We then feed $\mathcal{P}$ along with $\mathbf{x}$ as the multimodal instruction prompt to the MLLM to obtain $\mathcal{C}$. Figure 2 provides a sample of one such prompt template. Please refer Appendix C for more samples.

$\mathcal{C}$ provides a detailed description of $\mathbf{x}$, capturing the scene attributes specified by $\mathbf{g}$, such as object relationships, their interactions, or other spatial attributes. We pass $\mathcal{C}$ through a CLIP text encoder

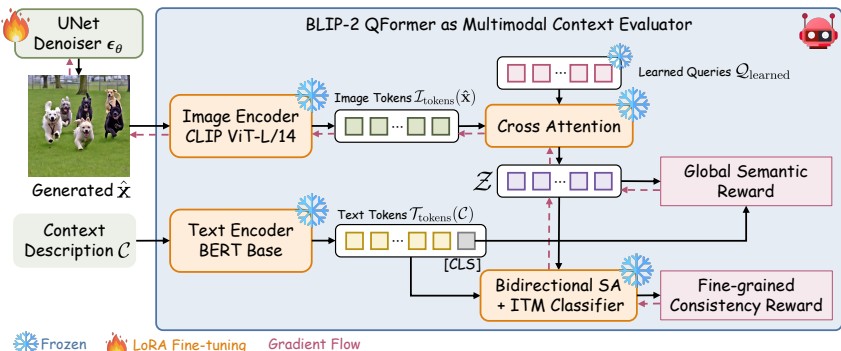

Figure 3: Hummingbird's Multimodal Context Evaluator leverages pre-trained BLIP-2 QFormer. It simultaneously maximizes our novel Global Semantic and Fine-grained Consistency Rewards to align generated image $\hat{\mathbf{x}}$ with Content Description $\mathcal{C}$ corresponding to multimodal context $\mathcal{M}$.

(Radford et al., 2021) to convert it into a text embedding $\mathcal{T}_{\mathbf{e}}$. In parallel, we also encode $\mathbf{x}$ using a CLIP image encoder to generate the image embedding $\mathcal{I}_{\mathbf{e}}$ comprising the semantic features of $\mathbf{x}$.

We feed the embeddings $\mathcal{I}_{\mathbf{e}}$ and $\mathcal{T}_{\mathbf{e}}$ to the UNet denoiser $\epsilon_\theta$ of SDXL (Podell et al., 2024) to generate image $\hat{\mathbf{x}}$. While $\hat{\mathbf{x}}$ exhibits diversity (i.e. visually different from $\mathbf{x}$), we fine-tune the denoiser via LoRA (Hu et al., 2022) to enhance the fidelity of $\hat{\mathbf{x}}$ (i.e. preserve the scene attributes occurring in $\mathbf{x}$ in relation to $\mathbf{g}$). To facilitate the fine-tuning, we introduce Multimodal Context Evaluator which simultaneously maximizes our two novel rewards – Global Semantic and Fine-Grained Consistency Rewards – to align $\hat{\mathbf{x}}$ with the scene attributes provided in the multimodal context $\mathcal{M}$ (Figure 3).

## 4.2 FINE-TUNING WITH MULTIMODAL CONTEXT REWARDS

**Multimodal Context Evaluator.** We design our Multimodal Context Evaluator using the multimodal representation connector in pre-trained vision-language models (VLMs). Specifically, we leverage the pre-trained QFormer from BLIP-2 (Li et al., 2023) due to its unique ability to capture both global and local alignment between image-text pairs. We use BLIP-2's image encoder to extract the image tokens $\mathcal{I}_{\text{tokens}}(\hat{\mathbf{x}})$ for the image $\hat{\mathbf{x}}$ generated by SDXL (Figure 3). In parallel, we use BLIP-2's BERT-based (Kenton & Toutanova, 2019) text encoder to extract the text tokens $\mathcal{T}_{\text{tokens}}$ (including the representative token $\mathcal{T}_{\text{[CLS]}}$) for the text-based Context Description $\mathcal{C}$ obtained from the MLLM. We then utilize the learned queries $\mathcal{Q}_{\text{learned}}$ from the pre-trained BLIP-2 QFormer to extract the visual representation token sequence $\mathcal{Z}$ via the Cross-Attention layer as,

$$\mathcal{Z} = \texttt{CrossAttention}\left(\mathcal{Q}_{\text{learned}}, \mathcal{I}_{\text{tokens}}(\hat{\mathbf{x}})\right) \tag{3}$$

**Global Semantic Reward.** Using the visual representation tokens $\mathcal{Z}$ and the representative text token $\mathcal{T}_{\text{[CLS]}}$, we compute cosine similarity between each visual token $\mathcal{Z}_i$ and $\mathcal{T}_{\text{[CLS]}}$ and define Global Semantic Reward, $\mathcal{R}_{\text{global}}$, as the maximum of these computed cosine similarities,

$$\mathcal{R}_{\text{global}} = \max_i \frac{\mathcal{Z}_i^T \cdot \mathcal{T}_{\text{[CLS]}}}{\|\mathcal{Z}_i\|\|\mathcal{T}_{\text{[CLS]}}\|} \tag{4}$$

Through the above formulation, the Global Semantic Reward, $\mathcal{R}_{\text{global}}$, ensures the overall scene context in the generated image $\hat{\mathbf{x}}$ aligns with that in the textual context description $\mathcal{C}$ by maximizing their global feature similarity.

**Fine-Grained Consistency Reward.** While $\mathcal{R}_{\text{global}}$ allows to align the global semantics of the generated image $\hat{\mathbf{x}}$ with that of the textual context description $\mathcal{C}$, it is not sufficient to maintain optimal fidelity as SDXL can still omit capturing key scene attributes specified in $\mathbf{g}$. We therefore introduce Fine-Grained Consistency Reward, $\mathcal{R}_{\text{fine-grained}}$, that helps to complement $\mathcal{R}_{\text{global}}$ and captures the multimodal context in $\hat{\mathbf{x}}$ more comprehensively in a fine-grained manner. For this, we leverage the bi-directional self-attention along with the image-text matching (ITM) classifier of pre-trained BLIP-2 QFormer to attend to all the visual tokens $\mathcal{Z}$ and Context Description text tokens $\mathcal{T}_{\text{tokens}}$ with each other. The ITM classifier outputs two logits: one for the positive match (with index $j = 1$) and one for the negative match (with index $j = 0$). In the training of Hummingbird, positive pairs are defined as the generated image and its corresponding context description within the same training

---

**Algorithm 1** Multimodal Context Rewards Fine-tuning

---

**Require:** Pre-trained UNet denoiser $\epsilon_\theta$; original data distribution $\mathbb{P}_{\text{data}}$; context descriptor MLLM.
**Ensure:** $\epsilon_\theta$ converges and minimizes $\mathcal{L}_{\text{total}}$.
    Activate UNet denoiser $\epsilon_\theta$; freeze context rewards $\mathcal{R}_{\text{global}}$, $\mathcal{R}_{\text{fine-grained}}$, context descriptor MLLM
    **while** $\mathcal{L}_{\text{total}}$ not converged **do**
        Sample multimodal context input $\{\mathbf{x}, \mathbf{g}\} \sim \mathbb{P}_{\text{data}}$; $t = T$
        $\mathcal{P} \leftarrow p(\mathbf{g})$                 ▷ Create instruction prompt $\mathcal{P}$ from text guidance $\mathbf{g}$ and prompt template $p$
        $\mathcal{C} \leftarrow \text{MLLM}(\mathbf{x}, \mathcal{P})$                     ▷ Extract context description $\mathcal{C}$ from MLLM
        Extract image embedding $\mathcal{I}_\mathbf{e}$ of reference image $\mathbf{x}$
        Extract text embedding $\mathcal{T}_\mathbf{e}$ of description $\mathcal{C}$
        **while** $t > 0$ **do**                               ▷ perform DDIM denoising
            $\mathbf{z}_{t-1} \leftarrow \sqrt{\alpha_{t-1}} \frac{\mathbf{z}_t - \sqrt{1-\alpha_t} \epsilon_\theta(\mathbf{z}_t, \mathcal{I}_\mathbf{e}, \mathcal{T}_\mathbf{e}, t)}{\sqrt{\alpha_t}} + \sqrt{1 - \alpha_{t-1}} \cdot \epsilon_\theta(\mathbf{z}_t, \mathcal{I}_\mathbf{e}, \mathcal{T}_\mathbf{e}, t)$
        **end while**
        $\hat{\mathbf{x}} \leftarrow \mathcal{D}(\mathbf{z}_0)$                  ▷ VAE Decoder decodes latent $\mathbf{z}_0$ to pixel space
        $\mathcal{L}_{\text{total}} \leftarrow -(\lambda_1 \mathcal{R}_{\text{global}} + \lambda_2 \mathcal{R}_{\text{fine-grained}})$
        Backward $\mathcal{L}_{\text{total}}$ and update $\epsilon_\theta$ for last $K$ steps.
    **end while**

---

batch. We select the logit corresponding to the positive match $j = 1$ of the classifier as $\mathcal{R}_{\text{fine-grained}}$,

$$\mathcal{R}_{\text{fine-grained}} = \text{ITM\_Classifier}(\text{Bidirectional\_SA}(\mathcal{Z}, \mathcal{T}_{\text{tokens}}))_{\text{j=1}}, \tag{5}$$

$\mathcal{R}_{\text{fine-grained}}$ therefore captures the fine-grained multimodal image-text alignment between $\hat{\mathbf{x}}$ and $\mathcal{C}$.

**Loss Function.** We combine $\mathcal{R}_{\text{global}}$ and $\mathcal{R}_{\text{fine-grained}}$ to compute loss $\mathcal{L}_{\text{total}}$ to fine-tune denoiser $\epsilon_\theta$,

$$\mathcal{L}_{\text{total}} = -(\lambda_1 \mathcal{R}_{\text{global}} + \lambda_2 \mathcal{R}_{\text{fine-grained}}), \tag{6}$$

where $\lambda_1$ and $\lambda_2$ are the hyperparameters balancing the contribution from the two rewards.

**Training Procedure.** During training, we freeze all components of Hummingbird except UNet denoiser $\epsilon_\theta$ in SDXL which we fine-tune using LoRA (Figure 2 and 3). Additionally, text guidance $\mathbf{g}$ comprises the ground truth corresponding to the reference image in relation to the input text query. Algorithm 1 presents an overview of the training procedure. We optimize the UNet denoiser $\epsilon_\theta$ in SDXL using DDIM (Song et al., 2021) scheduler, guided by the two rewards $\mathcal{R}_{\text{global}}$ and $\mathcal{R}_{\text{fine-grained}}$. For each training step, we extract the image embedding $\mathcal{I}_\mathbf{e}$ of the reference image and text embedding $\mathcal{T}_\mathbf{e}$ of context description $\mathcal{C}$ to condition the generation process. We obtain the generated image $\hat{\mathbf{x}}$ via 25-steps DDIM and VAE decoder $\mathcal{D}$ from latent $\mathbf{z}_0$ to pixel space $\hat{\mathbf{x}}$, which we feed to Multimodal Context Evaluator along with $\mathcal{C}$. Since each timestep in the denoising process is differentiable, we compute the gradient to update parameters $\theta$ in denoiser $\epsilon_\theta$ through chain rule,

$$\frac{\partial \mathcal{L}}{\partial \theta} = -\frac{\partial \mathcal{R}}{\partial \hat{\mathbf{x}}} \cdot \frac{\partial \hat{\mathbf{x}}}{\partial \mathbf{z}_0} \cdot \prod_{t=0}^{T} \frac{\partial \left[ \sqrt{\alpha_{t-1}} \frac{\mathbf{z}_t - \sqrt{1-\alpha_t} \epsilon_\theta(\mathbf{z}_t, \mathcal{I}_\mathbf{e}, \mathcal{T}_\mathbf{e}, t)}{\sqrt{\alpha_t}} + \sqrt{1 - \alpha_{t-1}} \cdot \epsilon_\theta(\mathbf{z}_t, \mathcal{I}_\mathbf{e}, \mathcal{T}_\mathbf{e}, t) \right]}{\partial \theta} \tag{7}$$

**Evaluation.** During evaluation, we remove the Multimodal Context Evaluator with final output being the generated image $\hat{\mathbf{x}}$ (Appendix D, Figure 9). For fair evaluation, text guidance $\mathbf{g}$ no longer includes the ground truth corresponding to the reference image in relation to the input text query.

## 5 EXPERIMENT

### 5.1 BENCHMARK FORMULATION

To evaluate Hummingbird's effectiveness as a multimodal context-aligned image generator, we introduce a benchmark focused on two key criteria: *fidelity*, which measures how accurately the generated images preserve scene attributes as per the multimodal context, and *diversity*, which assesses how distinct the generate images are from each other as well as from the reference image.

For **fidelity**, we build our evaluation framework around three main considerations: *Applicability*, ensuring generated images enhance model performance when combined with real data; *Efficiency*, emphasizing computationally light and resource-efficient protocols; and *Fairness*, promoting standardized test-time evaluation to mitigate biases caused by varying training processes.

Based on these principles, we adopt two evaluation settings: (1) VQA Benchmark for MLLMs using Test-Time Augmentation (TTA) (Shanmugam et al., 2021), and (2) Human-Object Interaction (HOI)

Reasoning using Test-time Prompt Tuning (TPT) from Shu et al. (2022). Both settings leverage real and synthetic data to boost performance while allowing computational efficiency. TTA uses pre-trained MLLMs without requiring additional training for fair comparison, while TPT fine-tunes prompt embeddings for quick convergence.

For **diversity**, we conduct two experiments using CLIP ViT-G/14 (Radford et al., 2021) features. First, we compute the Euclidean distance between the generated and reference images to quantify how distinct the generated images are from their reference counterparts. Second, we generate images using 20 different random seeds and compute the average pairwise Euclidean distance across all generated images, providing a measure of intra-set diversity.

**Datasets and Metrics.** For the VQA benchmark, we fine-tune Hummingbird on VQAv2 (Goyal et al., 2017) and GQA (Hudson & Manning, 2019), then evaluate using TTA on MME Perception (Fu et al., 2024), a common benchmark for assessing SOTA MLLMs. Our benchmark covers MME Perception tasks related to Existence, Count, Position, Color, and Scene (more discussion on this in Appendix A). We generate synthetic images using the test image as a reference and pair them with corresponding questions as text guidance. We then feed the real (reference) and generated images to MLLMs along with yes/no questions to extract the logit for the next token `[yes]`/`[no]`. We determine the final predicted token by averaging the logits and comparing it to the ground truth (yes/no). Using the paired yes and no questions for each test image, we report Accuracy (ACC), which measures the correctness of individual question predictions, and Accuracy+ (ACC+), which measures the joint correctness of the question pair.

For HOI Reasoning, we fine-tune Hummingbird on Bongard-HOI (Jiang et al., 2022) training set and evaluate on associated test sets using TPT from Shu et al. (2022). Following the setup, given a query test image with support sets of positive (e.g., *riding bicycle*) and negative (e.g., *not riding bicycle*) images, we generate synthetic images for support set images as reference using corresponding label as text guidance. We then use the augmented support sets to optimize a pair of prompt embeddings that contrast each other and predict the human-object interaction in query image by comparing its feature similarity to the two optimized prompts. Please refer to Shu et al. (2022) for more details. We use Accuracy as a measure of model's ability to correctly predict these human-object interactions.

**Method Comparisons.** We compare Hummingbird against representative techniques from four groups of SOTA image generation techniques (covered in Section 2): (1) customized T2I diffusion models (referred to as I2T2I SDXL), (2) Image Variation, (3) Image Translation/Editing, and (4) Textual Inversion, as well as data augmentation methods like RandAugment (Cubuk et al., 2020).

**Object-Centric Benchmark.** In addition to scene-aware tasks like VQA and HOI Reasoning, we also evaluate Hummingbird on an object-centric benchmark to demonstrate its versatility. Specifically, we fine-tune the UNet denoiser on the ImageNet training set (Deng et al., 2009), and perform TPT (Shu et al., 2022) using real and generated images on the ImageNet test set and four out-of-distribution (OOD) datasets: ImageNet-A (Hendrycks et al., 2021b), ImageNet-V2 (Recht et al., 2019), ImageNet-R (Hendrycks et al., 2021a), and ImageNet-Sketch (Wang et al., 2019). This enables us to assess the robustness of Hummingbird under natural distribution shifts. We use Top-1 accuracy which measures the correctness of classifying test images.

### 5.2 IMPLEMENTATION DETAILS

For Hummingbird, we use SDXL Base 1.0 which is a standard pre-trained diffusion-based image generation model. We further employ CLIP ViT-G/14 as the image encoder and both CLIP-L/14 & CLIP-G/14 as the text encoders (Radford et al., 2021). We perform LoRA fine-tuning with 11M trainable parameters ($\approx 0.46\%$ of total 2.6B parameters) on 8 NVIDIA A100 80GB GPUs using AdamW (Loshchilov & Hutter, 2019) optimizer, learning rate of `5e-6`, and gradient accumulation steps of 8. Please refer to Appendix Q for more details.

### 5.3 COMPARISON WITH EXISTING METHODS ON THE BENCHMARK FORMULATION

**VQA benchmark.** To evaluate Hummingbird for VQA using MME Perception, we experiment with SOTA MLLMs including LLaVA v1.6 (Liu et al., 2024) and InternVL 2.0 (Chen et al., 2024). Table 1 shows that Hummingbird significantly improves both ACC and ACC+ compared to all existing methods on both LLaVA v1.6 and InternVL 2.0 consistently across all tasks. The ability of Hummingbird's generated images to enhance MLLM performance on complex VQA tasks validates

Table 1: Comparison for VQA benchmark on MME Perception using Test-Time Augmentation (TTA). Hummingbird outperforms SOTA image generation and augmentation techniques consistently across all MME tasks, when evaluating with different MLLMs.

| MLLM Name | Method | Existence | | Count | | Position | | Color | | Scene | |
|---|---|---|---|---|---|---|---|---|---|---|---|
| | | ACC | ACC+ | ACC | ACC+ | ACC | ACC+ | ACC | ACC+ | ACC | ACC+ |
| LLaVA v1.6 7B (Liu et al., 2024) | Real only | 95.00 | 90.00 | 81.67 | 66.67 | 76.67 | 53.33 | 93.33 | 86.67 | 86.00 | 72.50 |
| | Real + RandAugment (Cubuk et al., 2020) | 93.33 ↓1.67 | 86.67 ↓3.33 | 78.33 ↓3.34 | 60.00 ↓6.67 | 78.33 ↑1.66 | 56.67 ↑3.34 | 91.67 ↓1.66 | 83.33 ↓3.34 | 85.50 ↓0.50 | 72.00 ↓0.50 |
| | Real + Image Variation (Xu et al., 2023) | 88.33 ↓6.67 | 76.67 ↓13.33 | 71.67 ↓10.00 | 46.67 ↓20.00 | 76.67 - | 56.67 ↑3.34 | 85.00 ↓8.33 | 73.33 ↓13.34 | 86.50 ↑0.50 | 73.00 ↑0.50 |
| | Real + Image Translation (Luzi et al., 2024) | 93.33 ↓1.67 | 86.67 ↓3.33 | 78.33 ↓3.34 | 60.00 ↓6.67 | 80.00 ↑3.33 | 60.00 ↑6.67 | 93.33 - | 86.67 - | 86.75 ↑0.75 | 73.00 ↑0.50 |
| | Real + Textual Inversion (Gal et al., 2023) | 86.67 ↓8.33 | 73.33 ↓16.67 | 71.67 ↓10.00 | 43.33 ↓23.34 | 73.33 ↓3.34 | 50.00 ↓3.33 | 85.00 ↓8.33 | 70.00 ↓16.67 | 85.25 ↓0.75 | 71.00 ↓1.50 |
| | Real + I2T2I SDXL (Podell et al., 2024) | 96.67 ↑1.67 | 93.33 ↑3.33 | 81.67 - | 66.67 - | 75.00 ↓1.67 | 50.00 ↓3.33 | 88.33 ↓5.00 | 76.67 ↓10.00 | 86.50 ↓0.50 | 72.50 - |
| | Real + **Hummingbird** | **96.67** ↑1.67 | **93.33** ↑3.33 | **83.33** ↑1.66 | **70.00** ↑3.33 | **81.67** ↑5.00 | **66.67** ↑13.34 | **95.00** ↑1.67 | **93.33** ↑6.66 | **87.75** ↑1.75 | **74.00** ↑1.50 |
| InternVL 2.0 8B (Chen et al., 2024) | Real only | 96.67 | 93.33 | 73.33 | 50.00 | 76.67 | 60.00 | 91.67 | 83.33 | 85.00 | 70.00 |
| | Real + RandAugment (Cubuk et al., 2020) | 95.00 ↓1.67 | 90.00 ↓3.33 | 76.67 ↑3.34 | 66.67 ↑16.67 | 76.67 - | 60.00 - | 88.33 ↓3.34 | 76.67 ↓6.66 | 84.75 ↓0.25 | 69.50 ↓0.50 |
| | Real + Image Variation (Xu et al., 2023) | 91.67 ↓5.00 | 83.33 ↓10.00 | 73.33 - | 53.33 ↑3.33 | 70.00 ↓6.67 | 46.67 ↓13.33 | 78.33 ↓13.34 | 60.00 ↓23.33 | 85.25 ↑0.25 | 70.50 ↑0.50 |
| | Real + Image Translation (Luzi et al., 2024) | 85.00 ↓11.67 | 70.00 ↓23.33 | 75.00 ↑1.67 | 50.00 - | 78.33 ↑1.66 | 63.33 ↑3.33 | 90.00 ↓1.67 | 80.00 ↓3.33 | 85.50 ↑0.50 | 70.50 ↑0.50 |
| | Real + Textual Inversion (Gal et al., 2023) | 83.33 ↓13.34 | 66.67 ↓26.66 | 70.00 ↓3.33 | 46.67 ↓3.33 | 61.67 ↓15.00 | 40.00 ↓20.00 | 75.00 ↓16.67 | 56.67 ↓26.66 | 84.25 ↓0.75 | 68.50 ↓1.50 |
| | Real + I2T2I SDXL (Podell et al., 2024) | 93.33 ↓3.34 | 86.67 ↓6.66 | 78.33 ↑5.00 | 56.67 ↑6.67 | 65.00 ↓11.67 | 43.33 ↓16.67 | 95.00 ↑3.33 | 90.00 ↑6.67 | 84.75 ↓0.25 | 70.50 ↑0.50 |
| | Real + **Hummingbird** | **98.33** ↑1.66 | **96.67** ↑3.34 | **86.67** ↑13.34 | **73.33** ↑23.33 | **78.33** ↑1.66 | **63.33** ↑3.33 | **98.33** ↑6.66 | **96.67** ↑13.34 | **86.25** ↑1.25 | **71.00** ↑1.00 |

Table 2: Comparison on Human-Object Interaction (HOI) Reasoning using Test-time Prompt Tuning (TPT). Hummingbird outperforms SOTA methods on all 4 test splits of Bongard-HOI dataset.

| Method | Test Splits | | | | Average |
|---|---|---|---|---|---|
| | seen act., seen obj. | unseen act., seen obj. | seen act., unseen obj. | unseen act., unseen obj. | |
| RandAugment (Cubuk et al., 2020) | 66.39 | 68.50 | 65.98 | 65.48 | 66.59 |
| Image Variation (Xu et al., 2023) | 57.86 ↓8.53 | 59.71 ↓8.79 | 56.07 ↓9.91 | 55.58 ↓9.90 | 57.31 ↓9.28 |
| Image Translation (Luzi et al., 2024) | 66.22 ↓0.17 | 68.13 ↓0.37 | 66.09 ↑0.11 | 66.12 ↑0.64 | 66.64 ↑0.05 |
| Textual Inversion (Gal et al., 2023) | 55.18 ↓11.21 | 59.16 ↓9.34 | 55.30 ↓10.68 | 54.16 ↓11.32 | 55.95 ↓10.64 |
| I2T2I SDXL (Podell et al., 2024) | 67.26 ↑0.87 | 69.25 ↑0.75 | 67.23 ↑1.25 | 65.76 ↑0.28 | 67.38 ↑0.79 |
| **Hummingbird** | **68.14** ↑1.75 | **70.95** ↑2.45 | **68.28** ↑2.30 | **67.56** ↑2.08 | **68.73** ↑2.14 |

that Hummingbird can generate images faithful to the input multimodal context. The improvement is particularly prominent for tasks needing a fine-grained understanding of the scene as in Count and Position. For LLaVA v1.6, Hummingbird boosts Position ACC by 5% and ACC+ by 13.34%, while for InternVL 2.0, it significantly improves both Count ACC and Color ACC+ by 13.34% and Count ACC+ by 23.33%. This also highlights Hummingbird's generalizability across different MLLMs.

**HOI Reasoning.** We experiment with CLIP-ResNet50 (Radford et al., 2021) to evaluate Hummingbird on Bongard-HOI using TPT. Table 2 shows that Hummingbird outperforms all existing augmentation/image generation methods consistently across all test splits, achieving the highest average accuracy of 68.73% (+2.14% over data augmentation-based baseline). While VQA on MME Perception helps to demonstrate Hummingbird's effectiveness on spatial scene attributes, Hummingbird's superior performance on Bongard-HOI demonstrates the method's ability to generate high-fidelity augmentations also for novel interaction-based scene attributes.

**Object-Centric benchmarks.** Table 3 shows the evaluation of Hummingbird on object-centric benchmarks, including ImageNet and its OOD variants (ImageNet-A, ImageNet-V2, ImageNet-R, and ImageNet-Sketch) using TPT. We can observe that Hummingbird consistently outperforms all other image generation/augmentation methods across all object-centric benchmarks. This validates Hummingbird's versatility by being effective on both scene-aware and object-centric tasks.

**Qualitative Study.** Figure 4 provides qualitative comparison between Hummingbird and other image generation methods across different scene-aware tasks from MME Perception and HOI Reasoning benchmarks. Hummingbird consistently surpasses all other approaches, achieving higher fidelity w.r.t. the input multimodal context (reference image along with text guidance) while main-

Table 3: Hummingbird outperforms SOTA generation/augmentation methods on object-centric datasets including out-of-distribution (OOD) variants showing its robustness to distribution shifts.

| Method | ImageNet | ImageNet-A | ImageNet-V2 | ImageNet-R | ImageNet-Sk. | Average | OOD Avg. |
|---|---|---|---|---|---|---|---|
| Real only | 58.10 | 22.81 | 53.00 | 53.90 | 33.50 | 42.26 | 40.80 |
| Real + RandAugment | 59.40 ↑ 1.30 | 27.34 ↑ 4.53 | 55.20 ↑ 2.20 | 56.80 ↑ 2.90 | 34.50 ↑ 1.00 | 46.65 ↑ 4.39 | 43.46 ↑ 2.66 |
| Real + Image Variation | 60.80 ↑ 2.70 | 31.06 ↑ 8.25 | 55.80 ↑ 2.80 | 58.80 ↑ 4.90 | 37.10 ↑ 3.60 | 48.71 ↑ 6.45 | 45.69 ↑ 4.89 |
| Real + Image Translation | 61.90 ↑ 3.80 | 32.14 ↑ 9.33 | 56.20 ↑ 3.20 | 59.60 ↑ 5.70 | 37.20 ↑ 3.70 | 49.41 ↑ 7.15 | 46.29 ↑ 5.49 |
| Real + Textual Inversion | 60.10 ↑ 3.00 | 30.85 ↑ 8.04 | 55.50 ↑ 2.50 | 57.40 ↑ 3.50 | 35.50 ↑ 2.00 | 47.87 ↑ 5.61 | 44.81 ↑ 4.01 |
| Real + I2T2I SDXL | 61.20 ↑ 3.10 | 31.56 ↑ 8.75 | 56.20 ↑ 3.20 | 58.50 ↑ 4.60 | 38.10 ↑ 4.60 | 49.11 ↑ 6.85 | 46.09 ↑ 5.29 |
| Real + **Hummingbird** | **62.60 ↑ 4.50** | **32.85 ↑ 10.04** | **56.50 ↑ 3.50** | **60.20 ↑ 6.30** | **38.80 ↑ 5.30** | **50.19 ↑ 7.93** | **47.09 ↑ 6.29** |

Figure 4: Generated image comparison between Hummingbird and SOTA methods on MME Perception and HOI Reasoning. Hummingbird achieves highest fidelity while maintaining high diversity.

Table 4: Euclidean distance (diversity) between features of real and generated images, and among generated images with varying random seeds. Higher is better. **Bold** denotes best and underline is second best. While achieving best fidelity (Table 1-3), Hummingbird provides second best diversity.

| Comparison Type | RandAugment | I2T2I SDXL | Image Variation | Image Translation | Textual Inversion | Hummingbird ✗ fine-tuning | Hummingbird ✓ fine-tuning |
|---|---|---|---|---|---|---|---|
| **Reference (real) vs generated** | 15.80 | **37.22** | 36.37 | 21.89 | 36.84 | 36.10 | 36.94 |
| **Among generated** | 18.51 | **27.14** | 25.85 | 20.39 | 26.46 | 26.08 | 26.67 |

taining high diversity w.r.t the reference image. For example, in Figure 4 (Row 1), the reference image depicts 4 people, and Hummingbird successfully maintains this count in the generated image. In contrast, I2T2I SDXL, Image Translation, and Textual Inversion generate images with 6, 5, and 2 people, respectively. Image Variation struggles even with image quality, with an estimated people count ranging between 3 and 7. Please refer to Appendix R for more qualitative results.

## 5.4 ABLATION STUDY

**Diversity Analysis.** Table 4 shows that Hummingbird (after fine-tuning) achieves the second-highest Euclidean distance score, both w.r.t the reference image and among generated images, slightly behind I2T2I SDXL, while securing the highest fidelity as shown in Tables 1 and 2. Figure 5 illustrates Hummingbird's high diversity in generated images across different random seeds, validating its ability to provide high fidelity w.r.t multimodal context while preserving diversity.

**Effectiveness of Multimodal Context Rewards.** Table 5 shows an ablation to evaluate the impact of Global Semantic $\mathcal{R}_{global}$ and Fine-Grained Consistency Reward $\mathcal{R}_{fine-grained}$ of Hummingbird. We use LLaVA 1.6 7B for evaluation while considering both LLaVA 1.6 7B and InternVL 2.0 8B as MLLM for Context Description. Table 5 (Row 1 and 5) show that Hummingbird achieves the best performance when both rewards are applied. Performance reduces in absence of either reward, especially on tasks requiring detailed multimodal context preservation (e.g. for Position and Count).

**Impact of MLLM as Context Descriptor.** Table 5 also helps to analyze the effect of using different MLLMs to obtain Context Description $\mathcal{C}$. We consider LLaVA 1.6 7B and InternVL 2.0

Table 5: Ablation on Multimodal Context Rewards, MLLM Context Descriptor, and Fine-tuning. **Bold** is overall best, blue is best per Context Descriptor, gray is baseline without fine-tuning.

| MLLM Name | Context Descriptor | Reward $\mathcal{R}_{global}$ | $\mathcal{R}_{fine\text{-}grained}$ | Existence ACC | ACC+ | Count ACC | ACC+ | Position ACC | ACC+ | Color ACC | ACC+ | Scene ACC | ACC+ |
|---|---|---|---|---|---|---|---|---|---|---|---|---|---|
| LLaVA v1.6 7B | LLaVA v1.6 7B (Liu et al., 2024) | ✓ | ✓ | 96.67 | 93.33 | 83.33 | **70.00** | 81.67 | 66.67 | 95.00 | 93.33 | 87.75 | 74.00 |
| | | ✗ | ✓ | 96.67 | 93.33 | 83.33 | **70.00** | 80.00 | 63.33 | 95.00 | 93.33 | 87.25 | 73.50 |
| | | ✓ | ✗ | 96.67 | 93.33 | 81.67 | 66.67 | 81.67 | **66.67** | 93.33 | 90.00 | 87.50 | 74.00 |
| | | ✗ | ✗ | 96.67 | 93.33 | 81.67 | 66.67 | 78.33 | 56.67 | 90.00 | 83.33 | 86.75 | 73.50 |
| | InternVL 2.0 8B (Chen et al., 2024) | ✓ | ✓ | 98.33 | 96.67 | 85.00 | 70.00 | 80.00 | 63.33 | 95.00 | 93.33 | 87.25 | 73.50 |
| | | ✗ | ✓ | 98.33 | 96.67 | 85.00 | 70.00 | 78.33 | 60.00 | 95.00 | 93.33 | 87.00 | 73.00 |
| | | ✓ | ✗ | 98.33 | 96.67 | 83.33 | 66.67 | 78.33 | 60.00 | 91.67 | 90.00 | 86.75 | 73.00 |
| | | ✗ | ✗ | 96.67 | 93.33 | 81.67 | 66.67 | 76.67 | 56.67 | 90.00 | 86.67 | 86.50 | 72.50 |

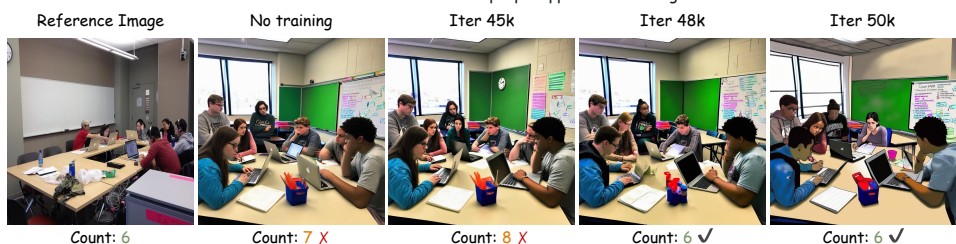

Reference Image | Seed 0 | Seed 1 | Seed 2 | Seed 3 | Seed 4

Text Guidance: Are there **three** giraffes? [Count / Ans: No]

Text Guidance: Is there a skateboard with red wheels? [Color / Ans: Yes]

Figure 5: Hummingbird exhibits high diversity across different random seeds while producing high-fidelity images each time w.r.t multimodal context (reference image + text guidance).

Text Guidance: Are there six people appear in this image?

Reference Image | No training | Iter 45k | Iter 48k | Iter 50k

Count: 6 | Count: 7 ✗ | Count: 8 ✗ | Count: 6 ✓ | Count: 6 ✓

Figure 6: Fine-tuning with Multimodal Context Rewards improves fidelity in generated images.

8B for evaluation. Table 5 (Row 1 and 5) show that training Hummingbird with $\mathcal{C}$ obtained from LLaVA performs better than that from InternVL for Position and Scene tasks while $\mathcal{C}$ from InternVL achieves better accuracy on Existence and Count tasks. This experiment emphasizes the importance of choosing a strong MLLM to get Context Description to fine-tune Hummingbird.

**Effectiveness of Fine-tuning.** Table 5 shows that fine-tuning SDXL in Hummingbird leads to better performance consistently across all tasks compared to the baseline without finetuning ( gray-shaded Rows 4, 8). Figure 6 further illustrates how fine-tuning improves fidelity while preserving diversity (with reference image having 6 people and text guidance referencing Count attribute, generated image also has 6 people after fine-tuning while remaining visually diverse w.r.t. reference image).

## 6 CONCLUSION

We introduce Hummingbird, a novel diffusion-based image generation algorithm that provides both high fidelity and diversity when generating images guided by multimodal context, consisting of a reference image and accompanying text guidance. By incorporating a Multimodal Context Evaluator and leveraging Global Semantic and Fine-Grained Consistency Rewards, Hummingbird ensures that the generated images accurately preserve specified scene attributes, such as object interactions and spatial relationships, while maintaining visual diversity. Our comprehensive experiments demonstrate that Hummingbird outperforms SOTA methods across multiple benchmarks, including VQA and HOI Reasoning, as well as object-centric benchmarks. The results validate Hummingbird's ability to generate high-fidelity, multimodal context-aligned images that improve scene-aware task performance while maintaining diversity of generated images. Hummingbird sets a new standard for scene-aware image generation, with potential applications for a wide range of multimodal tasks.

ACKNOWLEDGMENTS

This work was carried out during Minh-Quan's internship at Microsoft, and Dimitris Samaras was supported in part by NSF grants IIS-2123920 and IIS-2212046 during the preparation of the manuscript.

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

APPENDIX

## A  DISCUSSION: SCOPE AND ETHICS

In this work, we evaluate our method on six core scene-aware tasks: existence, count, position, color, scene, and HOI reasoning. We select these tasks as they represent core aspects of multimodal understanding which are essential for many applications. Meanwhile, we do not extend our evaluation to more complex reasoning tasks, such as numerical calculations or code generation, because SOTA diffusion models like SDXL are not yet capable of handling these tasks effectively. Fine-tuning alone cannot overcome the fundamental limitations of these models in generating images that require symbolic logic or complex reasoning. Additionally, we avoid tasks with ethical concerns, such as generating images of specific individuals (e.g., for celebrity recognition task), to mitigate risks related to privacy and misuse. Our goal was to ensure that our approach focuses on technically feasible and responsible AI applications. Expanding to other tasks will require significant advancements in diffusion model capabilities and careful consideration of ethical implications.

## B  LIMITATIONS AND FUTURE WORK

While our Multimodal Context Evaluator proves effective in enhancing the fidelity of generated images and maintaining diversity, Hummingbird is built using pre-trained diffusion models such as SDXL and MLLMs like LLaVA, it inherently shares the limitations of these foundation models. Hummingbird still faces challenges with complex reasoning tasks such as numerical calculations or code generation due to the symbolic logic limitations inherent to SDXL. Additionally, during inference, the MLLM context descriptor occasionally generates incorrect information or ambiguous descriptions initially, which can lead to lower fidelity in the generated images. Figure 7 further illustrates these observations.

Hummingbird currently focuses on single attributes like count, position, and color as part of the multimodal context. This is because this task alone poses significant challenges to existing methods, which Hummingbird effectively addresses. A potential direction for future work is to broaden the applicability of Hummingbird to synthesize images with multiple scene attributes in the multimodal context as part of compositional reasoning tasks.

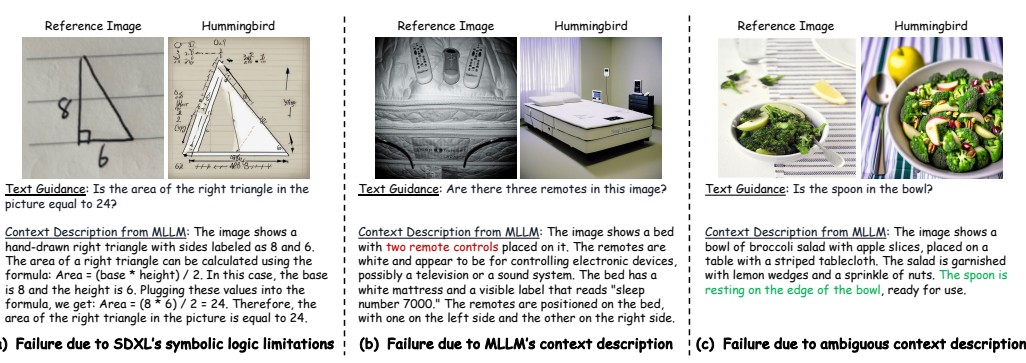

Figure 7: Failure cases of Hummingbird. (a) Our method fails due to the symbolic logic limitation of existing pre-trained SDXL. (b) Initially incorrect descriptions generated by MLLMs lead to low fidelity of generated images. (c) Context description generated by MLLMs is ambiguous and does not directly relate to the text guidance, the spoon can be both inside or outside the bowl.

## C  PROMPT TEMPLATES

Figure 8 (a-c) showcases the prompt templates used by Hummingbird to fine-tune diffusion models specifically on each task including VQA, HOI Reasoning, and Object-Centric benchmarks. It's worth noting that we designed the prompt such that it provides detailed instruction to MLLMs on

which scene attributes to focus. We also evaluate the effectiveness of our designed prompt templates by fine-tuning Hummingbird with a generic prompt as illustrated in Figure 8 (d). Table 6 indicates that without using our designed prompt template, the MLLM is not properly instructed to generate specific context description thus leading to reduced performance after fine-tuning on MME tasks. We believe that when using a generic prompt, MLLM is not able to receive sufficient grounding about the multimodal context leading to information loss on key scene attributes.

Table 6: Effectiveness of the prompt template on fine-tuning Hummingbird on MME Perception.

| MLLM Name | Hummingbird | Existence | | Count | | Position | | Color | | Scene | |
|---|---|---|---|---|---|---|---|---|---|---|---|
| | | ACC | ACC+ | ACC | ACC+ | ACC | ACC+ | ACC | ACC+ | ACC | ACC+ |
| **LLaVA v1.6 7B** (Liu et al., 2024) | w/ prompt template | **96.67** | **93.33** | **83.33** | **70.00** | **81.67** | **66.67** | **95.00** | **93.33** | **87.75** | **74.00** |
| | w/ generic prompt | 91.67 ↓5.00 | 83.33 ↓10.00 | 75.00 ↓8.33 | 56.67 ↓13.33 | 81.67 - | 63.33 ↓3.34 | 91.67 ↓3.33 | 83.33 ↓10.00 | 87.25 ↓0.50 | 73.00 ↓1.00 |
| **InternVL 2.0 8B** (Chen et al., 2024) | w/ prompt template | **98.33** | **96.67** | **86.67** | **73.33** | **78.33** | **63.33** | **98.33** | **96.67** | **86.25** | **71.00** |
| | w/ generic prompt | 91.67 ↓6.66 | 83.33 ↓13.34 | 80.00 ↓6.67 | 60.00 ↓13.33 | 71.67 ↓6.66 | 50.00 ↓13.33 | 91.67 ↓6.66 | 83.33 ↓13.34 | 84.50 ↓1.75 | 69.00 ↓2.00 |

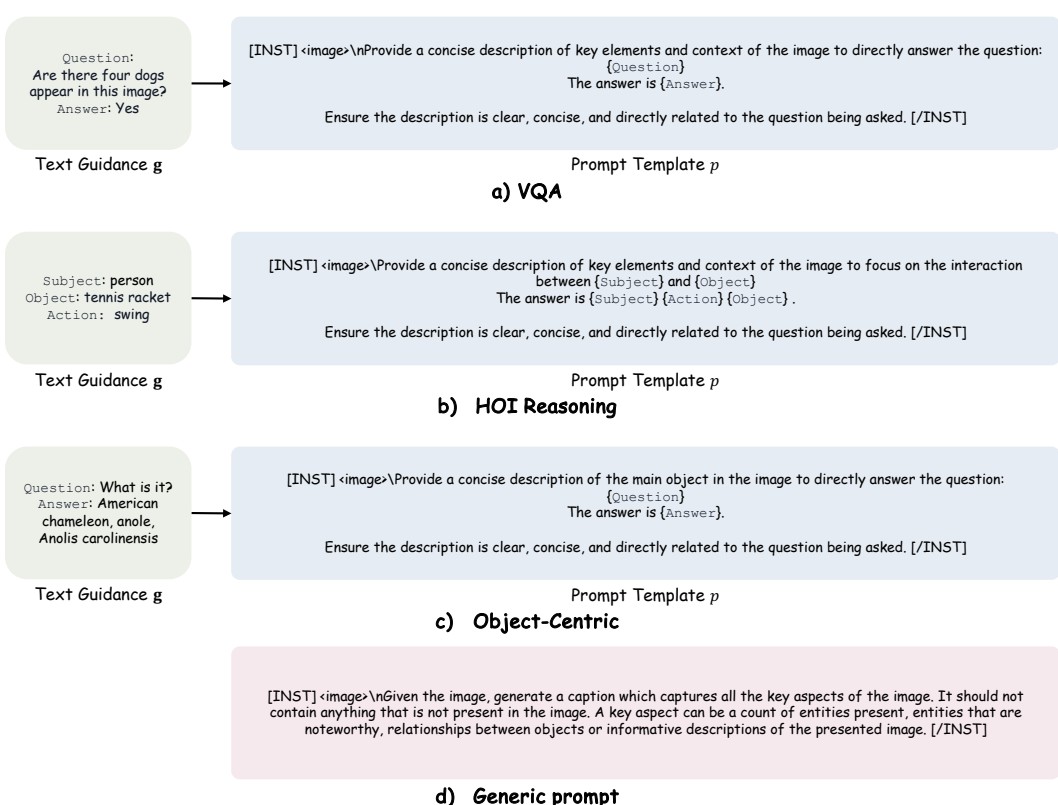

Figure 8: Prompt templates (a-c) used by Hummingbird to fine-tune the diffusion model on each task including VQA, HOI Reasoning, and Object Centric benchmarks. The generic prompt (d) is also included to evaluate the effectiveness of prompt template.

## D    INFERENCE PIPELINE

In the inference pipeline of Hummingbird (Figure 9), the text guidance **g** includes only the question corresponding to the reference image **x**. The answer is excluded for fair evaluation. Moreover, we remove Multimodal Context Evaluator, and the generated image **x̂** is the final output.

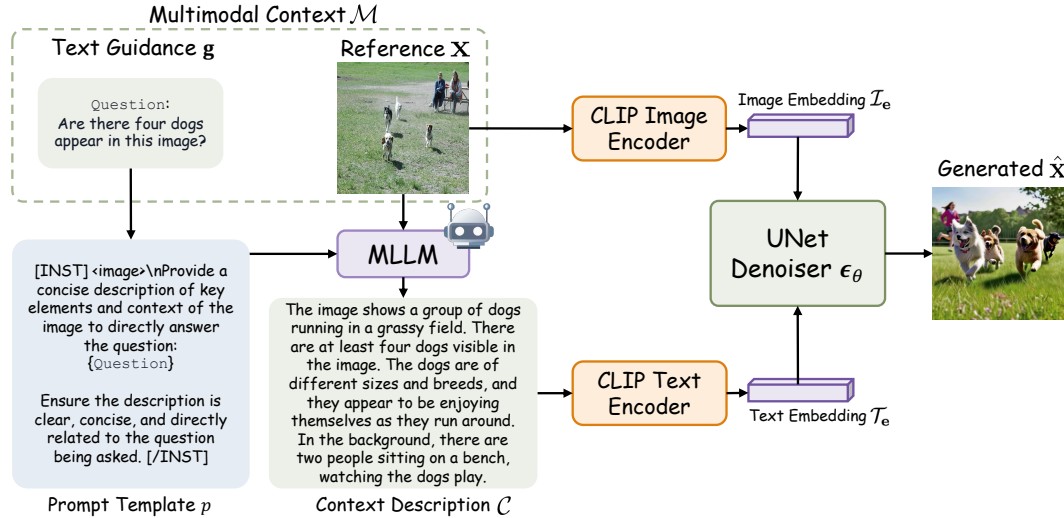

Figure 9: Inference pipeline of Hummingbird

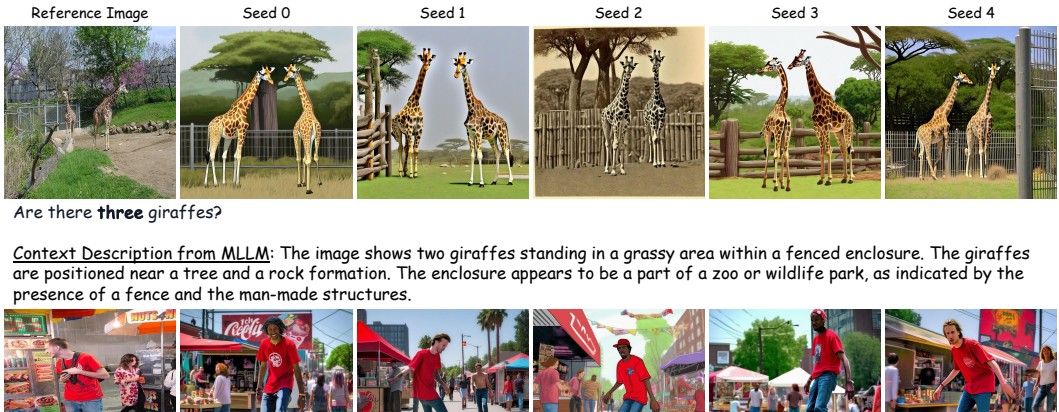

Figure 10: Examples of context description from MLLM in the inference pipeline where answers are not included in text guidance.

# E ABLATION STUDY ON BLIP-2 QFORMER

Our design choice to leverage BLIP-2 QFormer in Hummingbird as the multimodal context evaluator facilitates the formulation of our novel Global Semantic and Fine-grained Consistency Rewards. These rewards enable Hummingbird to be effective across all tasks as seen in Table 7. On replace with a less powerful multimodal context encoder such as CLIP ViT-G/14, we can only implement the global semantic reward as the cosine similarity between the text features and generated image features. As a result, while the setting can maintain performance on coarse-level tasks such as Scene and Existence, there is a noticeable decline on fine-grained tasks like Count and Position. This demonstrates the effectiveness of our design choices in Hummingbird and shows that using less

powerful alternatives, without the ability to provide both global and fine-grained alignment, affects the fidelity of generated images.

Table 7: Effectiveness of the prompt template on fine-tuning Hummingbird on MME Perception.

| MLLM Name | Hummingbird | Existence | | Count | | Position | | Color | | Scene | |
|---|---|---|---|---|---|---|---|---|---|---|---|
| | | ACC | ACC+ | ACC | ACC+ | ACC | ACC+ | ACC | ACC+ | ACC | ACC+ |
| **LLaVA v1.6 7B** (Liu et al., 2024) | w/ our Evaluator | **96.67** | **93.33** | **83.33** | **70.00** | **81.67** | **66.67** | **95.00** | **93.33** | **87.75** | **74.00** |
| | w/ CLIP | 96.67 - | 93.33 - | 81.67 ↓1.66 | 66.67 ↓3.33 | 80.00 ↓1.67 | 63.33 ↓3.34 | 95.00 - | 90.00 ↓3.33 | 87.75 - | 73.50 ↓0.50 |
| **InternVL 2.0 8B** (Chen et al., 2024) | w/ our Evaluator | **98.33** | **96.67** | **86.67** | **73.33** | **78.33** | **63.33** | **98.33** | **96.67** | **86.25** | **71.00** |
| | w/ CLIP | 98.33 - | 96.67 - | 81.67 ↓5.00 | 70.00 ↓3.33 | 76.67 ↓1.66 | 60.00 ↓3.33 | 96.67 ↓1.66 | 93.33 ↓3.34 | 86.00 ↓0.25 | 71.00 - |

# F  ADDITIONAL EVALUATION ON MME ARTWORK

To explore the method's ability to work on tasks involving more nuanced or abstract text guidance beyond factual scene attributes, we evaluate Hummingbird on an additional task of MME Artwork. This task focuses on image style attributes that are more nuanced/abstract such as the following question-answer pair – Question: "Does this artwork exist in the form of mosaic?", Answer: "No".

Table 8 summarizes the evaluation. We can observe that Hummingbird outperforms all existing methods on both ACC and ACC+, implying its higher effectiveness in generating images with high fidelity (in this case, image style preservation) compared to existing methods. This provides evidence that Hummingbird can generalize to tasks involving abstract/nuanced attributes such as image style. Figure 11 further shows qualitative comparison between image generation methods on the MME Artwork task.

Table 8: Comparison on Artwork benchmark and Visual Reasoning task. Hummingbird outperforms SOTA image generation and augmentation techniques.

| Method | Real only | RandAugment | Image Variation | Image Translation | Textual Inversion | I2T2I SDXL | Hummingbird |
|---|---|---|---|---|---|---|---|
| **Artwork ACC** | 69.50 | 69.25 | 69.00 | 67.00 | 66.75 | 68.00 | **70.25** |
| **Artwork ACC+** | 41.00 | 41.00 | 40.00 | 38.00 | 37.50 | 38.00 | **41.50** |
| **Reasoning ACC** | 69.29 | 67.86 | 69.29 | 69.29 | 67.14 | 72.14 | **72.86** |
| **Reasoning ACC+** | 42.86 | 40.00 | 41.40 | 40.00 | 37.14 | 47.14 | **48.57** |

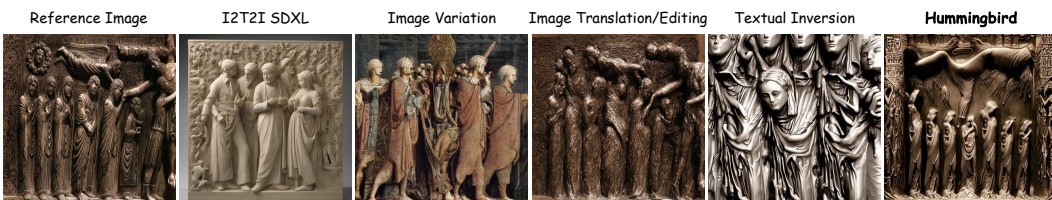

Text Guidance: Does this artwork exist in the form of sculpture?

Context Description from MLLM: The image depicts a relief sculpture, which is a form of art that is carved into a flat surface, such as a wall or a panel. The sculpture features a group of figures, including a woman and a man, who are interacting with each other. The figures are depicted in a realistic style, with attention to detail in their clothing and expressions.

Figure 11: Qualitative comparison on the Artwork task between image generation method. Hummingbird can preserve both diversity and fidelity of the reference image in a more abstract domain.

# G  ADDITIONAL EVALUATION ON MME COMMONSENSE REASONING

We have additionally performed our evaluation to more complex tasks such as Visual Reasoning using the MME Commonsense Reasoning benchmark. Results in Table 8 highlight Hummingbird's

ability to generalize effectively across diverse domains and complex reasoning tasks, demonstrating its broader applicability. Figure 12 further shows qualitative comparison between image generation methods on the MME Commonsense Reasoning task.

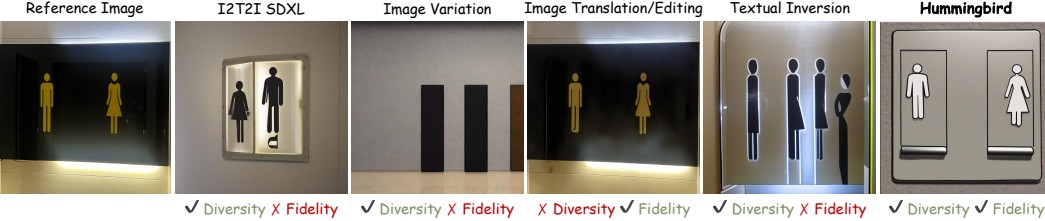

Text Guidance: This is a toilet guide sign. I am a man. Should I go to the toilet on the left?

Context Description from MLLM: The image shows a sign with a male and female symbol, indicating the presence of restrooms. The sign is mounted on a wall, and the male symbol is on the left side of the sign. The context suggests that the sign is providing guidance for restrooms, with the male symbol pointing to the left. Therefore, if you are a man, you should go to the toilet on the left.

Figure 12: Qualitative comparison on the Commonsense Reasoning task between image generation method. Hummingbird can preserve both diversity and fidelity of the reference image in a more abstract domain.

## H    FID SCORES

We compute FID scores for Hummingbird and the different baselines (traditional augmentation and image generation methods) and tabulate the numbers in Table 9. FID is a valuable metric for assessing the quality of generated images and how closely the distribution of generated images matches the real distribution. However, *FID does not account for the diversity among the generated images*, which is a critical aspect of the task our work targets (i.e., how can we generate high fidelity images, preserving certain scene attributes, while still maintaining high diversity?). We also illustrate the shortcomings of FID for the task in Figure 13 where we compare generated images across methods. We observe that RandAugment and Image Translation achieve lower FID scores than Hummingbird (w/ finetuning) because they compromise on diversity by only minimally changing the input image, allowing their generated image distribution to be much closer to the real distribution. While Hummingbird has a higher FID score than RandAugment and Image Translation, Figure 13 shows that it is able to preserve the scene attribute w.r.t. multimodal context while generating an image that is significantly different from than original input image. Therefore, it accomplishes the targeted task more effectively, with both high fidelity and high diversity.

Table 9: FID scores of traditional augmentation and image generation methods. Lower is better.

| Method | RandAugment | I2T2I SDXL | Image Variation | Image Translation | Textual Inversion | Hummingbird ✗ fine-tuning | ✓ fine-tuning |
|---|---|---|---|---|---|---|---|
| **FID score** ↓ | **15.93** | 18.35 | 17.66 | 16.29 | 20.84 | 17.78 | 16.55 |

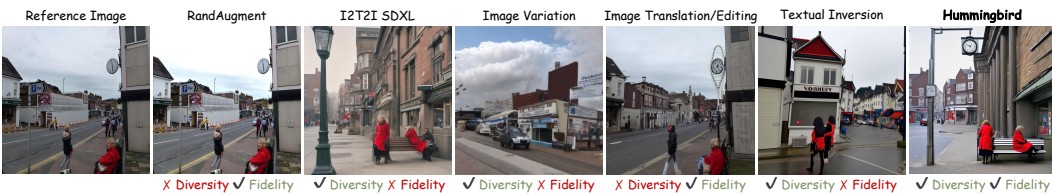

Figure 13: While RandAugment and Image Translation achieve lower FID scores, Hummingbird balances fidelity and diversity effectively.

## I    USER STUDY

We conduct a user study where we create a survey form with 50 questions (10 questions per MME Perception task). In each survey question, we show users a reference image, a related question, and a generated image each from two different methods (baseline I2T2I SDXL vs Hummingbird). We ask users to select the generated images(s) (either one or both or neither of them) that preserve the attribute referred to by the question in relation to the reference image. If an image is selected, it denotes high fidelity in generation. We collect form responses from 70 people for this study. We compute the percentage of total generated images for each method that were selected by the users as a measure of fidelity. Table 10 summarizes the results and shows that Hummingbird significantly outperforms I2T2I SDXL in terms of fidelity across all tasks on the MME Perception benchmark. We have some examples of survey questions in Figure 14.

Table 10: User study on MME tasks to evaluate the fidelity of generated images by I2T2I SDXL vs. Hummingbird.

| Method | Existence | Count | Position | Color | Scene | Average |
|---|---|---|---|---|---|---|
| **I2T2I SDXL** | 63.71 | 44.43 | 40.00 | 46.86 | 87.86 | 56.57 |
| **Hummingbird** | **81.29** | **72.29** | **59.57** | **77.14** | **90.00** | **76.06** |

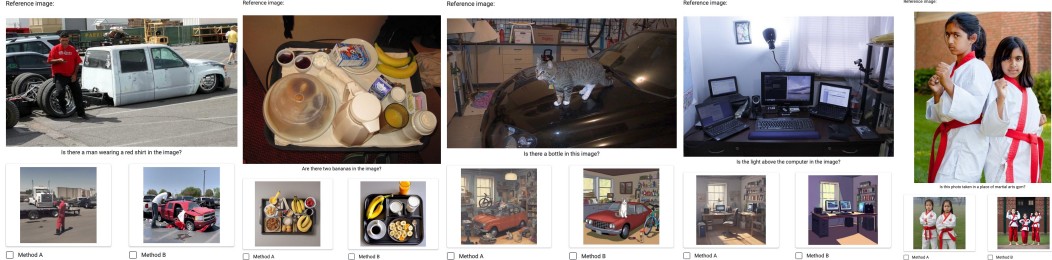

Figure 14: Some examples of our survey questions to evaluate the fidelity of generated images from I2T2I SDXL and Hummingbird.

## J    TRAINING PERFORMANCE ON BONGARD HOI DATASET

Following the existing method (Shu et al., 2022), we conduct an additional experiment by training a ResNet50 (He et al., 2016) model on the Bongard-HOI (Jiang et al., 2022) training set with traditional augmentation and Hummingbird generated images. We compare the performance with other image generation methods, using the same number of training iterations. As shown in Table 11, Hummingbird consistently outperforms all the baselines across all test splits. In the paper, as discussed in Section 5.1, we focus primarily on test-time evaluation because it eliminates the variability introduced by model training due to multiple external variables such as model architecture, data distribution, and training configurations, and allows for a fairer comparison where the evaluation setup remains fixed.

## K    RANDOM SEEDS SELECTION ANALYSIS

We conduct an additional experiment, varying the number of random seeds from 10 to 100. The results are presented in the boxplot in Figure 15, which shows the distribution of the mean L2 distances of generated image features from Hummingbird across different numbers of seeds.

The figure demonstrates that the difference in the distribution of the diversity (L2) scores across the different numbers of random seeds is statistically insignificant. So while it is helpful to increase the number of seeds for improved confidence, we observe that it stabilizes at 20 random seeds. This analysis suggests that using 20 random seeds also suffices to capture the diversity of generated images without significantly affecting the robustness of the analysis.

Table 11: Comparison on Human-Object Interaction (HOI) Reasoning by training a CNN-baseline ResNet50 with image augmentation and generation methods. Hummingbird outperforms SOTA methods on all 4 test splits of Bongard-HOI dataset.

| Method | Test Splits | | | | Average |
|---|---|---|---|---|---|
| | seen act., seen obj. | unseen act., seen obj. | seen act., unseen obj. | unseen act., unseen obj. | |
| CNN-baseline (ResNet50) | 50.03 | 49.89 | 49.77 | 50.01 | 49.92 |
| RandAugment (Cubuk et al., 2020) | 51.07 ↑ 1.04 | 51.14 ↑ 1.25 | 51.79 ↑ 2.02 | 51.73 ↑ 1.72 | 51.43 ↑ 1.51 |
| Image Variation (Xu et al., 2023) | 41.78 ↓ 8.25 | 41.29 ↓ 8.60 | 41.15 ↓ 8.62 | 41.25 ↓ 8.76 | 41.37 ↓ 8.55 |
| Image Translation (Luzi et al., 2024) | 46.60 ↓ 3.43 | 46.94 ↓ 2.95 | 46.38 ↓ 3.39 | 46.50 ↓ 3.51 | 46.61 ↓ 3.31 |
| Textual Inversion (Gal et al., 2023) | 37.67 ↓ 12.36 | 37.52 ↓ 12.37 | 38.12 ↓ 11.65 | 38.06 ↓ 11.95 | 37.84 ↓ 12.08 |
| I2T2I SDXL (Podell et al., 2024) | 51.92 ↑ 1.89 | 52.18 ↑ 2.29 | 52.25 ↑ 2.48 | 52.15 ↑ 2.14 | 52.13 ↑ 2.21 |
| **Hummingbird** | **53.71** ↑ 3.68 | **53.55** ↑ 3.66 | **53.69** ↑ 3.92 | **53.41** ↑ 3.40 | **53.59** ↑ 3.67 |

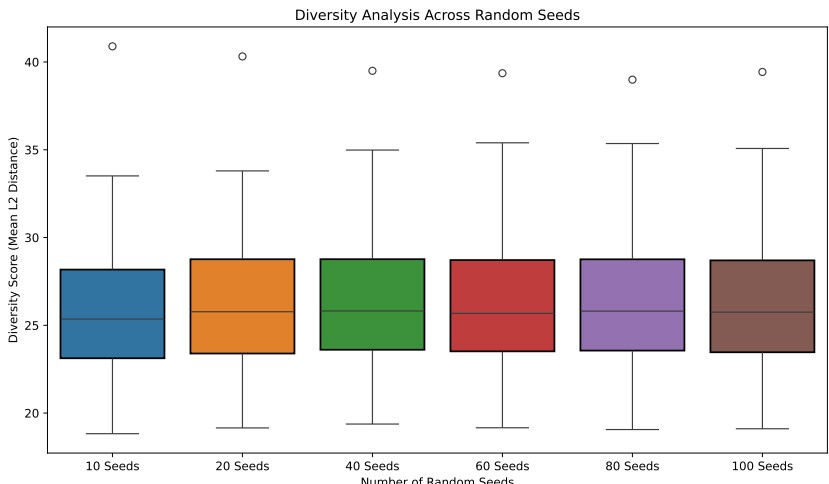

Figure 15: Diversity analysis across varying numbers of random seeds (10 to 100) using mean L2 distances of generated image features from Hummingbird. The box plot demonstrates consistent diversity scores as the number of seeds increases, indicating that performance stabilizes around 20 random seeds.

## L    FURTHER EXPLANATION OF MULTIMODAL CONTEXT EVALUATOR

The Global Semantic Reward, $\mathcal{R}_{\text{global}}$, ensures alignment between the global semantic features of the generated image $\hat{\mathbf{x}}$ and the textual context description $\mathcal{C}$. This reward leverages cosine similarity to measure the directional alignment between two feature vectors, which can be interpreted as maximizing the mutual information $I(\hat{\mathbf{x}}, \mathcal{C})$ between the generated image $\hat{\mathbf{x}}$ and the context description $\mathcal{C}$. Mutual information quantifies the dependency between the joint distribution $p_\theta(\hat{\mathbf{x}}, \mathcal{C})$ and the marginal distributions. In conditional diffusion models, the likelihood $p_\theta(\hat{\mathbf{x}}|\mathcal{C})$ of generating $\hat{\mathbf{x}}$ given $\mathcal{C}$ is proportional to the joint distribution:

$$p_\theta(\hat{\mathbf{x}}|\mathcal{C}) = \frac{p_\theta(\hat{\mathbf{x}}, \mathcal{C})}{p(\mathcal{C})} \propto p_\theta(\hat{\mathbf{x}}, \mathcal{C}),$$

where $p(\mathcal{C})$ is the marginal probability of the context description, treated as a constant during optimization. By maximizing $\mathcal{R}_{\text{global}}$, which aligns global semantic features, the model indirectly maximizes the mutual information $I(\hat{\mathbf{x}}, \mathcal{C})$, thereby enhancing the likelihood $p_\theta(\hat{\mathbf{x}}|\mathcal{C})$ in the conditional diffusion model.

The Fine-Grained Consistency Reward, $\mathcal{R}_{\text{fine-grained}}$, captures detailed multimodal alignment between the generated image $\hat{\mathbf{x}}$ and the textual context description $\mathcal{C}$. It operates at a token level, leveraging bidirectional self-attention and cross-attention mechanisms in the BLIP-2 QFormer, followed by the Image-Text Matching (ITM) classifier to maximize the alignment score.

**Self-Attention on Text Tokens:** Text tokens $\mathcal{T}_{\text{tokens}}$ undergo self-attention, allowing interactions among words to capture intra-text dependencies:

$$\mathcal{T}_{\text{attn}} = \texttt{SelfAttention}(\mathcal{T}_{\text{tokens}}) \tag{8}$$

**Self-Attention on Image Tokens:** Image tokens $\mathcal{Z}$ are derived from visual features of the generated image $\hat{\mathbf{x}}$ using a cross-attention mechanism:

$$\mathcal{Z} = \texttt{CrossAttention}(\mathcal{Q}_{\text{learned}}, \mathcal{I}_{\text{tokens}}(\hat{\mathbf{x}})) \tag{9}$$

These tokens then pass through self-attention to extract intra-image relationships:

$$\mathcal{Z}_{\text{attn}} = \texttt{SelfAttention}(\mathcal{Z}) \tag{10}$$

**Cross-Attention between Text and Image Tokens:** The text tokens $\mathcal{T}_{\text{attn}}$ and image tokens $\mathcal{Z}_{\text{attn}}$ interact through cross-attention to focus on multimodal correlations:

$$\mathcal{F} = \texttt{CrossAttention}(\mathcal{T}_{\text{attn}}, \mathcal{Z}_{\text{attn}}) \tag{11}$$

**ITM Classifier for Alignment:** The resulting multimodal features $\mathcal{F}$ are fed into the ITM classifier, which outputs two logits: one for positive match ($j = 1$) and one for negative match ($j = 0$). The positive class ($j = 1$) indicates strong alignment between the image-text pair, while the negative class ($j = 0$) indicates misalignment:

$$\mathcal{R}_{\text{fine-grained}} = \texttt{ITM\_Classifier}(\mathcal{F})_{\text{j=1}} \tag{12}$$

The ITM classifier predicts whether the generated image and the textual context description match. Maximizing the logit for the positive class $j = 1$ corresponds to maximizing the log probability $\log p(\hat{\mathbf{x}}, \mathcal{C})$ of the joint distribution of image and text. This process aligns the fine-grained details in $\hat{\mathbf{x}}$ with $\mathcal{C}$, increasing the mutual information between the generated image and the text features.

**Improving fine-grained relationships of CLIP.** While the CLIP Text Encoder, at times, struggles to accurately capture spatial features when processing longer sentences in the Multimodal Context Description, Hummingbird addresses this limitation by distilling the global semantic and fine-grained semantic rewards from BLIP-2 QFormer into a specific set of UNet denoiser layers, as mentioned in the implementation details under Appendix Q (i.e., Q, V transformation layers including `to_q`, `to_v`, `query`, `value`). This strengthens the alignment between the generated image tokens (Q) and input text tokens from the Multimodal Context Description (K, V) in the cross-attention mechanism of the UNet denoiser. As a result, we obtain generated images with improved fidelity, particularly w.r.t. spatial relationships, thereby helping to mitigate the shortcomings of vanilla CLIP Text Encoder in processing the long sentences of the Multimodal Context Description.

To illustrate further, a Context Description like "the dog under the pool" is processed in three steps: (1) self-attention is applied to the text tokens (K, V), enabling spatial terms like "dog," "under," and "pool" to interact; (2) self-attention is applied to visual features represented by the generated image tokens (Q) to extract intra-image relationships (3) cross-attention aligns this text features with visual features. The resulting alignment scores are used to compute the mean and select the positive class for the reward. Our approach to distill this reward into the cross-attention layers therefore ensures that spatial relationships and other fine-grained attributes are effectively captured, improving the fidelity of generated images.

## M  THE CHOICE OF TEXT ENCODER IN SDXL AND BLIP-2 QFORMER

The choice of text encoder in our pipeline is to leverage pre-trained models for their respective strengths. SDXL inherently uses the CLIP Text Encoder for its generative pipeline, as it is designed to process text embeddings aligned with the CLIP Image Encoder. In the Multimodal Context Evaluator, we use the BLIP-2 QFormer, which is pre-trained with a BERT-based text encoder.

## N  TEXTUAL INVERSION FOR DATA AUGMENTATION

In our experiments, we applied Textual Inversion for data augmentation as follows: given a reference image, Textual Inversion learns a new text embedding that captures the context of the reference

image (denoted as <context>). This embedding is then used to generate multiple augmented images by employing the prompt: "a photo of <context>". This approach allows Textual Inversion to create context-relevant augmentations for comparison in our experiments.

## O   CONVERGENCE CURVE

To evaluate convergence, we monitor the training process using the Global Semantic Reward and Fine-Grained Consistency Reward as criteria. Specifically, we observe the stabilization of these rewards over training iterations. Figure 16 presents the convergence curves for both rewards, illustrating their gradual increase followed by stabilization around 50k iterations. This steady state indicates that the model has learned to effectively align the generated images with the multimodal context.

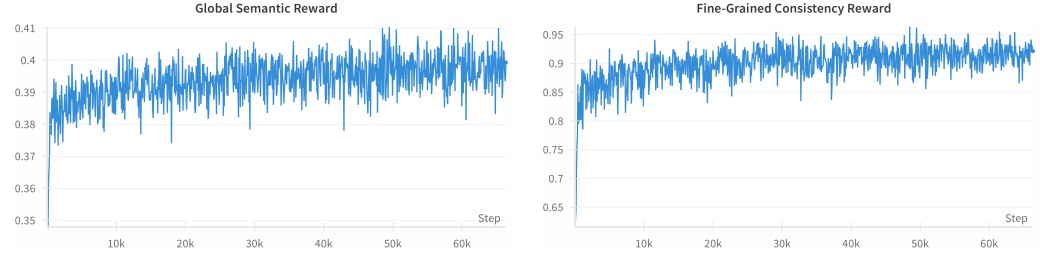

Figure 16: Convergence curves of Global Semantic and Fine-Grained Consistency Rewards

## P   FIDELITY EVALUATION USING GPT-4O

In addition to the results above, we compute additional metrics for fidelity, which measure how well the model preserves scene attributes when generating new images from a reference image. For this, we use GPT-4o (model version: 2024-05-13) as the MLLM oracle for a VQA task on the MME Perception benchmark (Fu et al., 2024). We evaluate Hummingbird with and without fine-tuning process.

The MME dataset consists of Yes/No questions, with a positive and a negative question for every reference image. To measure fidelity, we measure the rate at which the oracle's answer remains consistent across the reference and the generated image for every image in the dataset. We run the experiment multiple times and report the average numbers in Table 12. We see that fine-tuning the base SDXL with our novel rewards results in an average increase of 2.99% in fidelity.

Table 12: Fidelity between reference and generated images from Hummingbird with and without fine-tuning.

| MLLM Oracle | Hummingbird | Fidelity on "Yes" | Fidelity on "No" | Overall Fidelity |
|---|---|---|---|---|
| **GPT-4o** **Ver: 2024-05-13** | w/o fine-tuning | 68.33 | 70.55 | 71.18 |
| | w/ fine-tuning | **69.72** ↑ 1.39 | **73.61** ↑ 3.06 | **74.17** ↑ 2.99 |

## Q   IMPLEMENTATION DETAILS

We implement Hummingbird using PyTorch (Paszke et al., 2019) and HuggingFace diffusers (Face, 2023) libraries. For the generative model, we utilize the SDXL Base 1.0 which is a standard and commonly used pre-trained diffusion model in natural images domain. In the pipeline, we employ CLIP ViT-G/14 as image encoder and both CLIP-L/14 & CLIP-G/14 as text encoders (Radford et al., 2021). We perform LoRA fine-tuning on the following modules of SDXL UNet denoiser including $Q$, $V$ transformation layers, fully-connected layers (`to_q`, `to_v`, `query`, `value`, `ff.net.0.proj`)

with rank parameter $r = 8$, which results in 11M trainable parameters $\approx 0.46\%$ of total 2.6B parameters. The fine-tuning is done on 8 NVIDIA A100 80GB GPUs using AdamW (Loshchilov & Hutter, 2019) optimizer, a learning rate of `5e-6`, and gradient accumulation steps of 8.

## R    ADDITIONAL QUALITATIVE RESULTS

Figure 10 showcases two examples of context description from MLLM in the inference pipeline where answers are not included in text guidance. Figure 17 illustrates additional qualitative results highlighting the diversity and multimodal context fidelity between reference and synthetic images, as well as across images generated by Hummingbird with different random seeds. Figure 18 shows additional qualitative comparisons between Hummingbird and SOTA image generation methods on VQA and HOI Reasoning tasks.

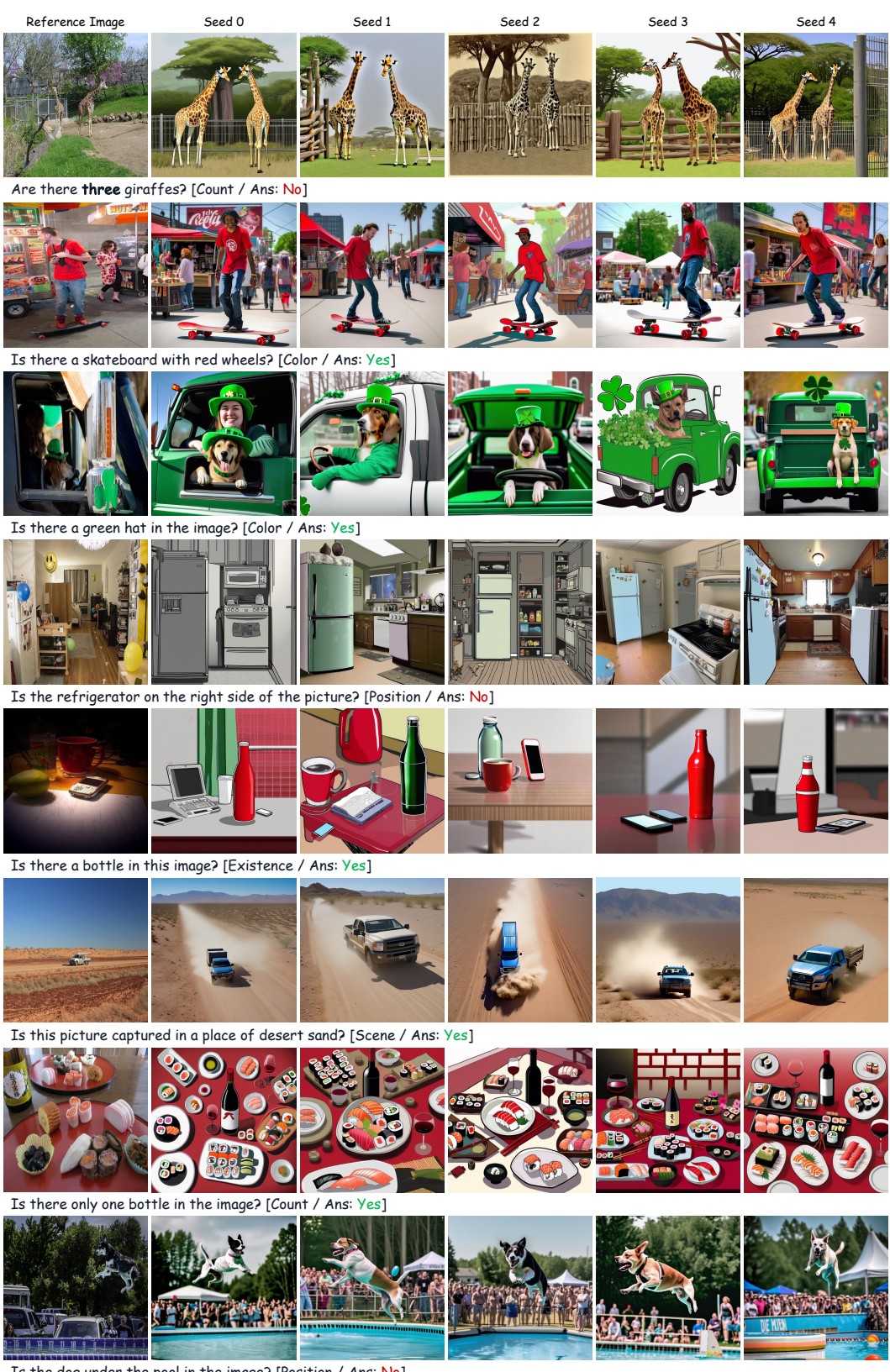

Figure 17: Diversity and multimodal context fidelity between reference and synthetic image and across generated ones from Hummingbird with different random seeds.

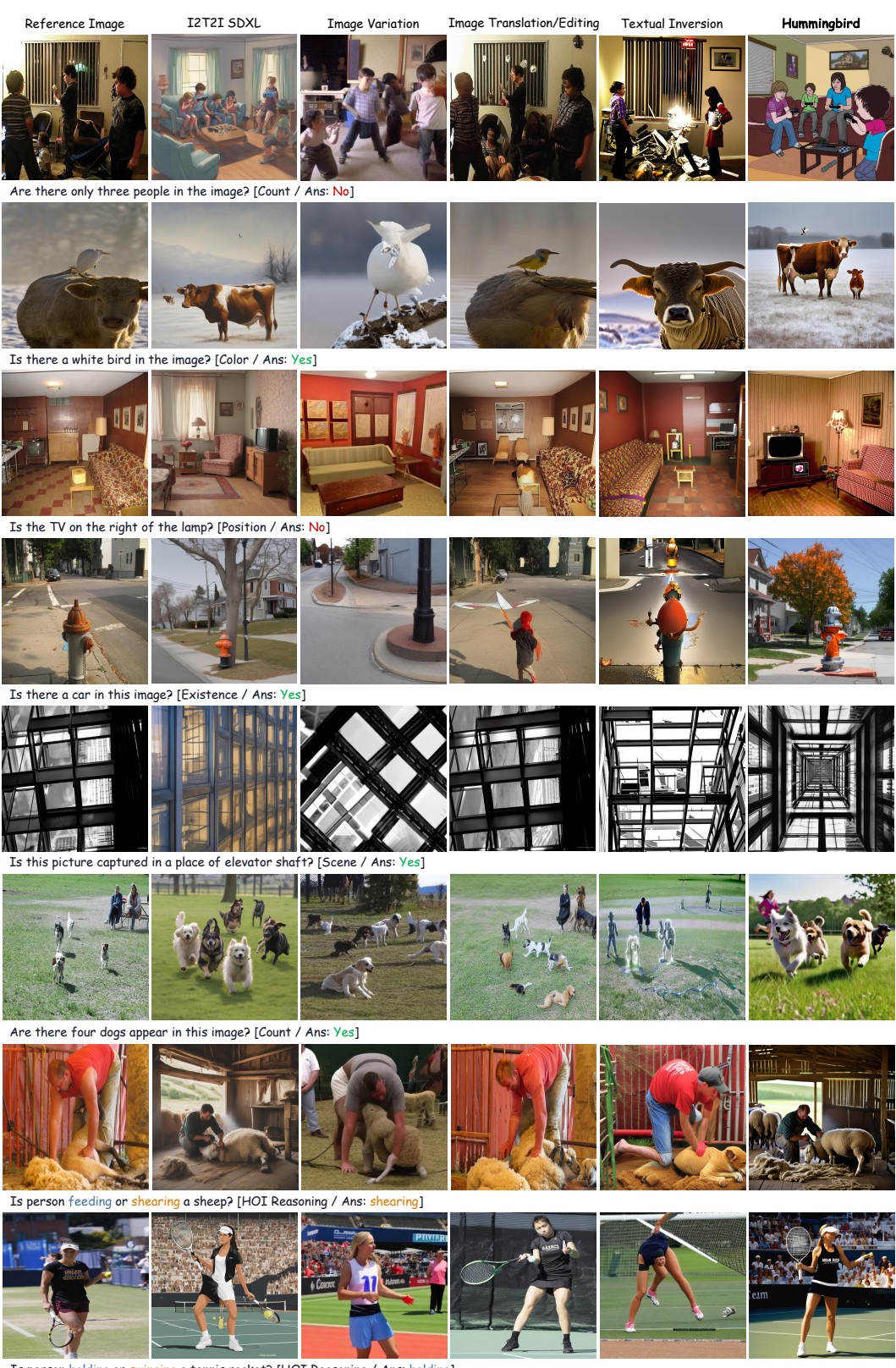

Figure 18: Qualitative comparison between Hummingbird and other image generation methods on MME Perception and HOI Reasoning benchmarks.

