# OpenReview forum: "Hummingbird: High Fidelity Image Generation via Multimodal Context Alignment"
_ICLR.cc/2025/Conference — ICLR 2025 Poster_

### Official Review · Reviewer_WWqb · 2024-10-27

**Soundness:** 3
**Presentation:** 3
**Contribution:** 3
**Rating:** 6
**Confidence:** 3

**Summary:**

The paper introduces Hummingbird, an image generation model that creates high-fidelity and diverse images aligned with multimodal context. It outperforms other methods on scene-aware tasks and uses a novel evaluator to optimize image generation.

**Strengths:**

1、High Fidelity and Diversity: Hummingbird generates images that are both diverse and maintain high fidelity to the multimodal context, which is crucial for complex visual tasks like VQA and HOI Reasoning.

2、Novel Multimodal Context Evaluator: The model uses a new evaluator that optimizes Global Semantic and Fine-grained Consistency Rewards, ensuring that generated images accurately preserve scene attributes from the reference image and text guidance.

3、Superior Performance: Benchmark experiments demonstrate that Hummingbird outperforms existing methods, showing its potential as a robust multimodal context-aligned image generator.

**Weaknesses:**

1、What is the use of using multimodal input as a condition? What are the benefits of using text as a condition compared to Stable Diffusion?

2、The sophisticated Multimodal Context Evaluator and the fine-tuning process might imply high computational requirements.

3、The performance of Hummingbird is likely to depend heavily on the quality and relevance of the multimodal context (reference image and text guidance) provided. In scenarios where the context is ambiguous or low-quality, the model's effectiveness may be compromised.

4、While Hummingbird shows strong performance on VQA and HOI Reasoning tasks, the document does not provide evidence of its effectiveness on a broader range of tasks.

**Questions:**

See weakness.

---

> ### Author Response · Authors · 2024-11-25
> **Response to Reviewer WWqb**
>
> We thank the reviewer for your thoughtful feedback. We address your comments below and have incorporated the feedback in main paper Introduction, and Appendix F, G.
>
>
> ---
> > **Q1 - What is the use of using multimodal input as a condition? What are the benefits of using text as a condition compared to Stable Diffusion?**
>
> Using multimodal input as a condition combines the strengths of both text and visual guidance, enabling richer scene understanding and more precise control over image generation. The reference image grounds the generation process in a clear visual context, while the text guidance specifies the attributes or relationships to focus on.
>
> We clarify that compared to Stable Diffusion, which relies solely on text prompts, our approach relies jointly on text and image prompts. This ensures better fidelity to scene attributes (e.g., object counts, spatial relationships) while maintaining high diversity and therefore allows supporting complex scene-aware tasks like VQA and HOI Reasoning.
>
> ---
> > **Q2 - The sophisticated Multimodal Context Evaluator and the fine-tuning process might imply high computational requirements.**
>
> We observe that fidelity in image generation primarily relies on the alignment in the cross-attention layers of the UNet denoiser. Therefore, we limited the fine-tuning process to these layers and employed LoRA, which reduces the number of trainable parameters to just 0.46% of the full UNet denoiser. This significantly minimizes the computational requirements while achieving performance gains up to 13.34% ACC and 23.23% ACC+ through augmentation via Hummingbird compared to using real images only.
>
>
> ---
> > **Q3 - The performance of Hummingbird is likely to depend heavily on the quality and relevance of the multimodal context (reference image and text guidance) provided. In scenarios where the context is ambiguous or low-quality, the model's effectiveness may be compromised.**
>
>
> While the presence of ambiguity or low quality of multimodal context has the potential to affect image generation, Hummingbird introduces multiple improvements to mitigate this. As shown in Table 6 in the Appendix, using generic prompts to transform the multimodal context into a context description can lead to reduced effectiveness in cases of ambiguity or low quality, resulting in a loss of fidelity on key scene attributes. In contrast, our designed prompt template helps to ground the context in entities of interest and provide task-specific instructions, enabling Hummingbird to demonstrate significantly higher performance across various tasks.
>
> ---
> > **Q4 - While Hummingbird shows strong performance on VQA and HOI Reasoning tasks, the document does not provide evidence of its effectiveness on a broader range of tasks.**
>
> We have additionally validated Hummingbird on more complex tasks such as Visual Reasoning using the MME Commonsense Reasoning benchmark as well as on more nuanced domains like image style on MME Artwork as suggested by Reviewer Ag2o. Results in the table below highlight Hummingbird's ability to generalize effectively across diverse domains and complex reasoning tasks, demonstrating its broader applicability. Please see Appendix F and G for qualitative results.
>
> Our additional experiment together with our performance on VQA and HOI Reasoning tasks show Hummingbird's effectiveness on a broader range of tasks. We also evaluated its robustness on object-centric datasets such as ImageNet and its OOD variants, including ImageNet-A, ImageNet-V2, ImageNet-R, and ImageNet-Sketch. As shown in Table 3 of the main paper, Hummingbird exhibits strong robustness to distribution shifts.
>
> | **Method**           | **Real only** | **RandAugment** | **Image Variation** | **Image Translation** | **Textual Inversion** | **I2T2I SDXL** | **Hummingbird** |
> |-----------------------|---------------|-----------------|---------------------|------------------------|------------------------|----------------|-------------|
> | **Artwork ACC**       | 69.50         | 69.25           | 69.00              | 67.00                 | 66.75                 | 68.00         | **70.25**   |
> | **Artwork ACC+**      | 41.00         | 41.00           | 40.00              | 38.00                 | 37.50                 | 38.00         | **41.50**   |
> | **Reasoning ACC**     | 69.29         | 67.86           | 69.29              | 69.29                 | 67.14                 | 72.14         | **72.86**   |
> | **Reasoning ACC+**    | 42.86         | 40.00           | 41.40              | 40.00                 | 37.14                 | 47.14         | **48.57**   |

---

### Official Review · Reviewer_spMH · 2024-10-31

**Soundness:** 3
**Presentation:** 3
**Contribution:** 4
**Rating:** 6
**Confidence:** 4

**Summary:**

The paper presents a new diffusion-based image generation method designed to address the challenge of maintaining both diversity and high fidelity in multimodal contexts. The main contributions are:

1. Introducing Hummingbird, a diffusion model capable of generating high-fidelity and diverse images based on multimodal context (a reference image and text guidance).

2. Proposing a novel Multimodal Context Evaluator that simultaneously maximizes global semantic and fine-grained consistency rewards, ensuring that the generated images maintain scene attributes from the multimodal context while preserving diversity.

3. Presenting a new benchmark using the MME Perception and Bongard HOI datasets, demonstrating Hummingbird's superiority in generating high-fidelity and diverse images compared to existing methods.

**Strengths:**

Originality: The paper introduces a new multimodal context alignment approach that balances diversity and fidelity. The introduction of a Multimodal Context Evaluator and reward mechanism demonstrates high originality.

Quality: The experimental design is well-conducted, clearly validating the proposed method's effectiveness in maintaining diversity and high fidelity.

Significance: Generating high-fidelity and diverse images is crucial for many complex visual tasks, particularly those involving scene understanding. Hummingbird demonstrates excellent performance in this area.

Clarity: The paper is well-organized, with a natural flow between sections, and the experimental results clearly highlight the comparative advantages over existing methods.

**Weaknesses:**

1. Lack of comprehensive theoretical basis: While global semantic and fine-grained consistency rewards are proposed, there is a lack of detailed mathematical derivation or theoretical analysis, especially regarding why these rewards are effective in improving fidelity.

2. Limited evaluation dataset diversity: The paper uses the MME and Bongard HOI datasets, but their representativeness may be limited, particularly regarding generalizing the model to broader scenarios. It is recommended to validate the method on more diverse datasets in future work.

**Questions:**

1. What is the basis for selecting the global semantic and fine-grained consistency rewards in the Multimodal Context Evaluator? Could more mathematical derivation or theoretical support be provided to explain the effectiveness of these reward mechanisms?

2. The experiments primarily use the MME and Bongard HOI datasets. Could the performance of the method be validated on larger or more diverse datasets? This would be crucial to demonstrate the generalizability of the method.

---

> ### Author Response · Authors · 2024-11-25
> **Response to Reviewer spMH (1/2)**
>
> We thank the reviewer for your thoughtful feedback. We address your comments below and have incorporated the feedback (main paper Introduction, Appendix F, G, L).
>
> ---
> > **Q1 - What is the basis for selecting the global semantic and fine-grained consistency rewards in the Multimodal Context Evaluator? Could more mathematical derivation or theoretical support be provided to explain the effectiveness of these reward mechanisms?**
>
>
> The Global Semantic Reward, $\mathcal{R}\_\textrm{global}$, ensures alignment between the global semantic features of the generated image $\mathbf{\hat{x}}$ and the textual context description $\mathcal{C}$. This reward leverages cosine similarity to measure the directional alignment between two feature vectors, which can be interpreted as maximizing the mutual information $I(\mathbf{\hat{x}}, \mathcal{C})$ between the generated image $\mathbf{\hat{x}}$ and the context description $\mathcal{C}$. Mutual information quantifies the dependency between the joint distribution $p_{\theta}(\mathbf{\hat{x}}, \mathcal{C})$ and the marginal distributions. In conditional diffusion models, the likelihood $p_{\theta}(\mathbf{\hat{x}} \vert \mathcal{C})$ of generating $\mathbf{\hat{x}}$ given $\mathcal{C}$ is proportional to the joint distribution:
>
> $p_{\theta}(\mathbf{\hat{x}} \vert \mathcal{C}) = \frac{p_{\theta}(\mathbf{\hat{x}}, \mathcal{C})}{p(\mathcal{C})} \propto p_{\theta}(\mathbf{\hat{x}}, \mathcal{C}),$
>
> where $p(\mathcal{C})$ is the marginal probability of the context description, treated as a constant during optimization. By maximizing $\mathcal{R}\_\textrm{global}$, which aligns global semantic features, the model indirectly maximizes the mutual information $I(\mathbf{\hat{x}}, \mathcal{C})$, thereby enhancing the likelihood $p_{\theta}(\mathbf{\hat{x}} \vert \mathcal{C})$ in the conditional diffusion model.
>
>
> The Fine-Grained Consistency Reward, $\mathcal{R}\_{\textrm{fine-grained}}$, captures detailed multimodal alignment between the generated image $\mathbf{\hat{x}}$ and the textual context description $\mathcal{C}$. It operates at a token level, leveraging bidirectional self-attention and cross-attention mechanisms in the BLIP-2 QFormer, followed by the Image-Text Matching (ITM) classifier to maximize the alignment score.
>
> **Self-Attention on Text Tokens:** Text tokens $\mathcal{T}\_{\mathrm{tokens}}$ undergo self-attention, allowing interactions among words to capture intra-text dependencies:
>
> $\mathcal{T}\_{\mathrm{attn}} = \tt{SelfAttention}(\mathcal{T}\_{\mathrm{tokens}})$
>
> **Self-Attention on Image Tokens:** Image tokens $\mathcal{Z}$ are derived from visual features of the generated image $\mathbf{\hat{x}}$ using a cross-attention mechanism:
>
> $\mathcal{Z} = \tt{CrossAttention}(\mathcal{Q}\_{\mathrm{learned}}, \mathcal{I}\_{\mathrm{tokens}}(\mathbf{\hat{x}}))$
>
> These tokens then pass through self-attention to extract intra-image relationships:
>
> $\mathcal{Z}\_{\mathrm{attn}} = \tt{SelfAttention}(\mathcal{Z})$
>
>
> **Cross-Attention between Text and Image Tokens:** The text tokens $\mathcal{T}\_{\mathrm{attn}}$ and image tokens $\mathcal{Z}\_{\mathrm{attn}}$ interact through cross-attention to focus on multimodal correlations:
>
> $\mathcal{F} = \tt{CrossAttention}(\mathcal{T}\_{\mathrm{attn}}, \mathcal{Z}\_{\mathrm{attn}})$
>
>
> **ITM Classifier for Alignment:** The resulting multimodal features $\mathcal{F}$ are fed into the ITM classifier, which outputs two logits: one for positive match ($j=1$) and one for negative match ($j=0$). The positive class ($j=1$) indicates strong alignment between the image-text pair, while the negative class ($j=0$) indicates misalignment:
>
> $\mathcal{R}\_{\textrm{fine-grained}} = \tt{ITM\\_Classifier}(\mathcal{F})\_{j=1}$
>
>
> The ITM classifier predicts whether the generated image and the textual context description match. Maximizing the logit for the positive class $j=1$ corresponds to maximizing the log probability $\log p(\mathbf{\hat{x}}, \mathcal{C})$ of the joint distribution of image and text. This process aligns the fine-grained details in $\mathbf{\hat{x}}$ with $\mathcal{C}$, increasing the mutual information between the generated image and the text features.

---

> > ### Author Response · Authors · 2024-11-25
> > **Response to Reviewer spMH (2/2)**
> >
> > > **Q2 - The experiments primarily use the MME and Bongard HOI datasets. Could the performance of the method be validated on larger or more diverse datasets? This would be crucial to demonstrate the generalizability of the method.**
> >
> > In addition to the MME and Bongard HOI datasets, we also conducted evaluations on object-centric datasets in the main paper (Table 3), including ImageNet, ImageNet-A, ImageNet-V2, ImageNet-R, and ImageNet-Sketch. These datasets provide a diverse range of evaluation scenarios: ImageNet contains over 1.2 million images spanning 1,000 classes, while its variants, such as ImageNet-A, include challenging adversarial examples, and ImageNet-Sketch focuses on stylized, sketch-like depictions of objects. As shown in Table 3 of the main paper, these experiments demonstrate the robustness of Hummingbird to distribution shifts and validate its ability to perform on larger/diverse datasets. We revised the introduction section of the manuscript to state it more clearly.
> >
> > Furthermore, we extend the evaluation of Hummingbird to more nuanced and abstract domains, such as image style (using the MME Artwork benchmark), and to a more complex task, Visual Reasoning (on the MME Commonsense Reasoning benchmark). Results in the table below confirm Hummingbird's generalization capability across diverse domains and its effectiveness in tackling more abstract and complex reasoning tasks. Please see Appendix F and G for qualitative results.
> >
> > | **Method**           | **Real only** | **RandAugment** | **Image Variation** | **Image Translation** | **Textual Inversion** | **I2T2I SDXL** | **Hummingbird** |
> > |-----------------------|---------------|-----------------|---------------------|------------------------|------------------------|----------------|-------------|
> > | **Artwork ACC**       | 69.50         | 69.25           | 69.00              | 67.00                 | 66.75                 | 68.00         | **70.25**   |
> > | **Artwork ACC+**      | 41.00         | 41.00           | 40.00              | 38.00                 | 37.50                 | 38.00         | **41.50**   |
> > | **Reasoning ACC**     | 69.29         | 67.86           | 69.29              | 69.29                 | 67.14                 | 72.14         | **72.86**   |
> > | **Reasoning ACC+**    | 42.86         | 40.00           | 41.40              | 40.00                 | 37.14                 | 47.14         | **48.57**   |

---

### Official Review · Reviewer_g5Ub · 2024-11-02

**Soundness:** 2
**Presentation:** 2
**Contribution:** 3
**Rating:** 6
**Confidence:** 4

**Summary:**

This paper proposes an image data augmentation pipeline based on diffusion models. Paired reference image and text guidance embeddings have been used into a diffusion model with LoRA to generate an image, and then the image can be optimized by a multimodal context evaluator who returns a global semantic reward and fine-grained consistency reward. Experimental results have been conducted to prove its effectiveness

**Strengths:**

1. The first work applying diffusion models for image data augmentation.
2. A pioneering study demonstrating the potential of synthetic data.
3. Produces impressive results.

**Weaknesses:**

1. The writing needs improvement; for example, the introduction should clearly state that the research task focuses on data augmentation.
2. Consider adding the following experiments: 1) evaluation of augmented image quality, such as using FID scores and user studies. 2) more assessment of the proposed augmentation's performance in training, not test-time. 3) Inclusion of a baseline in Table 4, such as "random seed + stable diffusion," to compare data augmentation capabilities, as the vanilla diffusion model does have variety, and I think 20 random seeds are not enough.
3. Other aspects mentioned in Questions.

**Questions:**

1. Could you provide further details on how to enhance the fidelity of generated images with respect to spatial relationships? While the CLIP Text Encoder is effective, it sometimes struggles to accurately capture spatial features when processing the longer sentences in the Context Description in Figure 2.
2. when generating the x_hat, you use CLIP Image Encoder and CLIP Text Encoder. However, in the BLIP-2 module, you opt for the BeRT text encoder instead. Could you clarify the rationale behind this choice?
3.  How is Textual Inversion, which fine-tunes a rarely used text embedding to learn novel concepts, being applied for data augmentation in your comparison experiments?
4.  Regarding line 274, what criteria do you use for convergence? Additionally, could you present your convergence curve in experiment?

---

> ### Author Response · Authors · 2024-11-25
> **Response to Reviewer g5Ub (1/3)**
>
> We thank the reviewer for your thoughtful feedback. We address your comments below and have incorporated the feedback (main paper Introduction, Appendix H, I, J, K, L, M, N, O).
>
> ---
>
> > **Q1 - The writing needs improvement; for example, the introduction should clearly state that the research task focuses on data augmentation.**
>
> While Hummingbird is effective for data augmentation, we will clarify in the introduction that it is in fact a general-purpose image generator. Moreover, Hummingbird is unique in its ability to leverage a multimodal context to generate images with both high fidelity and diversity. This makes Hummingbird broadly useful in many real-world scenarios that require both creativity and control (such as advertisement [1], e-commerce [2], art [3], etc.). Oftentimes, users find it challenging to put their imagination into words only. It is more convenient for them to illustrate their vision through a sample reference image along with text guidance on which attribute of the image they wish to preserve (such as the scene, number of objects, or spatial relationships) while letting the image generator perturb everything else. This necessitates a combination of fine-grained image understanding and high-fidelity image generation while still preserving the ability to generate with high diversity. Hummingbird is purposely designed to achieve this functionality.
>
> Reference:
>
> [1] Xue et al., "Strictly-ID-Preserved and Controllable Accessory Advertising Image Generation", arXiv 2024.
>
> [2] Chen et al., "VirtualModel: Generating Object-ID-retentive Human-object Interaction Image by Diffusion Model for E-commerce Marketing", arXiv 2024.
>
> [3] Jamwal and Ramaneswaran, "Composite Diffusion: whole>= $\Sigma$parts", WACV 2024.
>
> ---
>
> > **Q2 - Consider adding the following experiments: 1) FID scores and user studies. 2) the method's performance in training. 3) Vanilla diffusion 4) 20 random seeds are not enough.**
>
>
> We follow your recommendations and have added the following experiments in Appendix H, I, J, K:
>
> **FID scores.** We compute FID scores for Hummingbird and the different baselines (traditional augmentation and image generation methods) and tabulate the numbers in the table below. FID is a valuable metric for assessing the quality of generated images and how closely the distribution of generated images matches the real distribution. However, *FID does not account for the diversity among the generated images*, which is a critical aspect of the task our work targets (i.e., how can we generate high fidelity images, preserving certain scene attributes, while still maintaining high diversity?). We also illustrate the shortcomings of FID for the task in Figure 13 in the Appendix where we compare generated images across methods. We observe that RandAugment and Image Translation achieve lower FID scores than Hummingbird (w/ finetuning) because they compromise on diversity by only minimally changing the input image, allowing their generated image distribution to be much closer to the real distribution. While Hummingbird has a higher FID score than RandAugment and Image Translation, Figure 13 shows that it is able to preserve the scene attribute w.r.t. multimodal context while generating an image that is significantly different from than original input image. Therefore, it accomplishes the targeted task more effectively, with both high fidelity and high diversity.
>
> | **Method**              | **RandAugment** | **I2T2I SDXL** | **Image Variation** | **Image Translation** | **Textual Inversion** | **Hummingbird (w/o fine-tuning)** | **Hummingbird (w/ fine-tuning)** |
> |--------------------------|-----------------|----------------|---------------------|------------------------|------------------------|-----------------------------|-----------------------------|
> | **FID score↓**         | **15.93**       | 18.35          | 17.66              | 16.29                 | 20.84                 | 17.78                      | 16.55                      |

---

> ### Author Response · Authors · 2024-11-25
> **Response to Reviewer g5Ub (2/3)**
>
> **User study.** We conduct a user study where we create a survey form with 50 questions (10 questions per MME Perception task). In each survey question, we show users a reference image, a related question, and a generated image each from two different methods (baseline I2T2I SDXL vs Hummingbird). We ask users to select the generated images(s) (either one or both or neither of them) that preserve the attribute referred to by the question in relation to the reference image. If an image is selected, it denotes high fidelity in generation. We collect form responses from 70 people for this study. We compute the percentage of total generated images for each method that were selected by the users as a measure of fidelity. The table below summarizes the results and shows that Hummingbird significantly outperforms I2T2I SDXL in terms of fidelity across all tasks on the MME Perception benchmark. We have also added some examples of survey questions in Appendix I, Figure 14.
>
>
> | **Method**        | **Existence** | **Count** | **Position** | **Color** | **Scene** | **Average** |
> |--------------------|---------------|-----------|--------------|-----------|-----------|-------------|
> | **I2T2I SDXL**    | 63.71         | 44.43     | 40.00        | 46.86     | 87.86     | 56.57       |
> | **Hummingbird**       | **81.29**     | **72.29** | **59.57**    | **77.14** | **90.00** | **76.06**   |
>
> ---
> **The method's performance in training.** Following the existing method [4], we conduct an additional experiment by training a ResNet50 model on the Bongard-HOI training set with traditional augmentation and Hummingbird generated images. We compare the performance with other image generation methods, using the same
> number of training iterations. As shown in the table below, Hummingbird consistently outperforms all the baselines across all test splits. In the paper, as discussed in Section 5.1, we focus primarily on test-time evaluation because it eliminates the variability introduced by model training due to multiple external variables such as model architecture, data distribution, and training configurations, and allows for a fairer comparison where the evaluation setup remains fixed.
>
>
>
>
> | **Method**                  | **Seen act., seen obj.** | **Unseen act., seen obj.** | **Seen act., unseen obj.** | **Unseen act., unseen obj.** | **Average**            |
> |-----------------------------|--------------------------|----------------------------|----------------------------|-----------------------------|-------------------------|
> | CNN-baseline (ResNet50)     | 50.03                   | 49.89                     | 49.77                     | 50.01                      | 49.92                  |
> | RandAugment                 | 51.07 (+1.04)             | 51.14 (+1.25)               | 51.79 (+2.02)               | 51.73 (+1.72)                | 51.43 (+1.51)            |
> | Image Variation             | 41.78 (-8.25)             | 41.29 (-8.60)               | 41.15 (-8.62)               | 41.25 (-8.76)                | 41.37 (-8.55)            |
> | Image Translation           | 46.60 (-3.43)             | 46.94 (-2.95)               | 46.38 (-3.39)               | 46.50 (-3.51)                | 46.61 (-3.31)            |
> | Textual Inversion           | 37.67 (-12.36)            | 37.52 (-12.37)              | 38.12 (-11.65)              | 38.06 (-11.95)               | 37.84 (-12.08)           |
> | I2T2I SDXL                  | 51.92 (+1.89)             | 52.18 (+2.29)               | 52.25 (+2.48)               | 52.15 (+2.14)                | 52.13 (+2.21)            |
> | **Hummingbird**                 | **53.71 (+3.68)**         | **53.55 (+3.66)**           | **53.69 (+3.92)**           | **53.41 (+3.40)**            | **53.59 (+3.67)**        |
>
> Reference:
>
> [4] Shu et al., "Test-Time Prompt Tuning for Zero-Shot Generalization in Vision-Language Models", NeurIPS 2022.
>
> ---
> **Vanilla diffusion in diversity comparison in Table 4.** For the diversity comparison in Table 4, we focus on the methods compatible with the task our work targets where a method can process a multimodal context comprising input image and text. Moreover, standard image augmentation also requires a reference image to generate variations as augmentations. While vanilla stable diffusion can exhibit variety (diversity), it is a text-to-image model that does not include a reference input image and so we are unable to include it from comparison in the table. The closest baseline to vanilla diffusion is Image Translation where vanilla diffusion is modified to send reference image as input along with text guidance. We already included this baseline in Table 4 of main paper which we observe exhibits less diversity than Hummingbird.

---

> ### Author Response · Authors · 2024-11-25
> **Response to Reviewer g5Ub (3/3)**
>
> **20 random seeds are not enough.** We conduct an additional experiment where we vary the number of seeds from 10 to 100. We present the results as a boxplot in Appendix K, Figure 15 which shows the distribution of the mean L2 distances of generated image features from Hummingbird across different numbers of seeds.
>
> The figure demonstrates that the difference in the distribution of the diversity (L2) scores across the different numbers of random seeds is statistically insignificant. So while it is helpful to increase the number of seeds for improved confidence, we observe that it stabilizes at 20 random seeds. This analysis suggests that using 20 random seeds also suffices to capture the diversity of generated images without significantly affecting the robustness of the analysis.
>
>
> ---
>
> > **Q3 - Could you provide further details on how to enhance the fidelity of generated images with respect to spatial relationships? While the CLIP Text Encoder is effective, it sometimes struggles to accurately capture spatial features when processing the longer sentences in the Context Description in Figure 2.**
>
> While the CLIP Text Encoder, at times, struggles to accurately capture spatial features when processing longer sentences in the Multimodal Context Description, Hummingbird addresses this limitation by distilling the global semantic and fine-grained semantic rewards from BLIP-2 QFormer into a specific set of UNet denoiser layers, as mentioned in the implementation details under Appendix Q (i.e., Q, V transformation layers including $\tt{to\\_q, to\\_v, query, value}$). This strengthens the alignment between the generated image tokens (Q) and input text tokens from the Multimodal Context Description (K, V) in the cross-attention mechanism of the UNet denoiser. As a result, we obtain generated images with improved fidelity, particularly w.r.t. spatial relationships, thereby mitigating the shortcomings of the vanilla CLIP Text Encoder in processing long sentences of the Multimodal Context Description.
>
> To illustrate further, a Context Description like “the dog under the pool” is processed in three steps: (1) self-attention is applied to the text tokens (K, V), enabling spatial terms like “dog,” “under,” and “pool” to interact; (2) self-attention is applied to visual features represented by the generated image tokens (Q) to extract intra-image relationships (3) cross-attention aligns this text features with visual features. The resulting alignment scores are used to compute the mean and select the positive class for the reward. Our approach to distill this reward into the cross-attention layers therefore ensures that spatial relationships and other fine-grained attributes are effectively captured, improving the fidelity of generated images.
>
> ---
> > **Q4 - when generating the $\hat{\mathbf{x}}$, you use CLIP Image Encoder and CLIP Text Encoder. However, in the BLIP-2 module, you opt for the BeRT text encoder instead. Could you clarify the rationale behind this choice?**
>
> The choice of text encoder in our pipeline is to leverage pre-trained models for their respective strengths. SDXL inherently uses the CLIP Text Encoder for its generative pipeline, as it is designed to process text embeddings aligned with the CLIP Image Encoder. In the Multimodal Context Evaluator, we use the BLIP-2 QFormer, which is pre-trained with a BERT-based text encoder.
>
> ---
> > **Q5 - How is Textual Inversion, which fine-tunes a rarely used text embedding to learn novel concepts, being applied for data augmentation in your comparison experiments?**
>
> In our experiments, we applied Textual Inversion for data augmentation as follows: given a reference image, Textual Inversion learns a new text embedding that captures the context of the reference image (denoted as $<$context$>$). This embedding is then used to generate multiple augmented images by employing the prompt: "a photo of $<$context$>$". This approach allows Textual Inversion to create context-relevant augmentations for comparison in our experiments.
>
> ---
> > **Q6 - Regarding line 274, what criteria do you use for convergence? Additionally, could you present your convergence curve in experiment?**
>
> To evaluate convergence, we monitor the training process using the Global Semantic Reward and Fine-Grained Consistency Reward as criteria. Specifically, we observe the stabilization of these rewards over training iterations. Figure 16 in Appendix O presents the convergence curves for both rewards, illustrating their gradual increase followed by stabilization around 50k iterations. This steady state indicates that the model has learned to effectively align the generated images with the multimodal context.

---

> > ### Comment · Reviewer_g5Ub · 2024-12-02
> > **Reply to Response**
> >
> > R5: Based on your description, I think Textual Inversion may not be appropriate as a baseline for data augmentation.
> > ****
> > I reviewed the feedback from other reviewers and the authors' responses and appreciate their efforts to enhance the work.
> >
> > Overall, most of my concerns have been addressed and confusion been clarified, and I still think the writing requires improvement.
> >
> > Thus I increase my score to 6.

---

> > > ### Author Response · Authors · 2024-12-04
> > > **Many thanks to Reviewer g5Ub**
> > >
> > > Thank you for your detailed review and for increasing your score, we greatly value your support and input.

---

### Official Review · Reviewer_Ag2o · 2024-11-04

**Soundness:** 4
**Presentation:** 4
**Contribution:** 3
**Rating:** 8
**Confidence:** 4

**Summary:**

The paper introduces Hummingbird, a diffusion-based image generator that aligns generated images with a multimodal context comprising a reference image and text guidance. The model combines Global Semantic and Fine-Grained Consistency Rewards by a Multimodal Context Evaluator, leveraging vision-language models (BLIP-2). Hummingbird generates high-fidelity images that preserve scene attributes while maintaining diversity, performing favorably against state-of-the-art (SOTA) methods in tasks such as Visual Question Answering (VQA) and Human-Object Interaction (HOI) Reasoning.

**Strengths:**

1.	Interesting framework. The use of the Multimodal Context Evaluator with reward mechanisms (Global Semantic and Fine-Grained Consistency) is a unique approach that successfully addresses both the fidelity and diversity.
2.	Comprehensive Evaluation. The model is tested across various benchmarks and datasets, including VQAv2, GQA, and ImageNet, validating robustness under both scene-aware and object-centric tasks.
3.	Performance Gains. Empirical results show that Hummingbird consistently performs favorably against the other SOTA methods in terms of accuracy and consistency for VQA and HOI tasks. This validates the effectiveness of the proposed method in downstream tasks.
4.	Detailed Analysis: The paper includes thorough ablation studies that explore the impact of individual components and different pretrained MLLMs.

**Weaknesses:**

1.	Clarity of the Fine-Grained Consistency Reward. How the ITM classifier's positive class is determined sholud be clarified further.  What does the class ‘j’ mean in equation (5)?
2.	Limitations are not discussed. It would be more insightful to discuss about the potential limitations and possible improvement of the idea.

**Questions:**

### Questions
1.	How does the ITM classifier select the positive class for computing the Fine-Grained Consistency Reward?
2.	Would the model maintain robust performance when using alternative, less powerful MLLMs or other multimodal context encoders in place of BLIP-2?
3.	Could the method be adapted for tasks involving more nuanced or abstract text guidance beyond factual scene attributes, such as visual structures (e.g., relative positioning of objects) or style?

### Comments
- Including failure cases or limitations would provide more completeness of the paper.
- The paper would give more insights if the paper could outline about the future work.

---

> ### Author Response · Authors · 2024-11-25
> **Response to Reviewer Ag2o (1/2)**
>
> We thank the reviewer for your thoughtful feedback. We address your comments below and have incorporated the feedback (main paper Subsection 4.2, Appendix B, E, F).
>
> ---
>
> > **Q1 - Clarity of the Fine-Grained Consistency Reward. How the ITM classifier's positive class is determined should be clarified further. What does the class ‘j’ mean in equation (5)? How does the ITM classifier select the positive class for computing the Fine-Grained Consistency Reward?**
>
> The Image-Text Matching (ITM) classifier in the pre-trained BLIP-2 QFormer outputs two logits: one for the positive match (with index $j=1$) and one for the negative match (with index $j=0$). In the training of Hummingbird, positive pairs are defined as the generated image and its corresponding context description within the same training batch, while negative pairs consist of the generated image and unrelated context descriptions from the same training batch.
>
> To compute the Fine-Grained Consistency Reward, we maximize the logit corresponding to the positive match ($j=1$) output by the ITM classifier. By doing so, we encourage stronger alignment between the generated image and its corresponding context description. This ensures that specific scene attributes referenced in the context description are preserved, helping the UNet denoiser to capture fine-grained details and maintain fidelity during image generation.
>
> ---
>
> > **Q2 - It would be more insightful to discuss the potential limitations and possible improvement of the idea. Including failure cases or limitations would provide more completeness of the paper. The paper would give more insights if the paper could outline future work.**
>
> We discussed the potential limitations of Hummingbird in Appendix A of the paper and have added more details in Appendix B. While our Multimodal Context Evaluator proves effectiveness in enhancing the fidelity of generated images and maintaining diversity, Hummingbird is built using pre-trained diffusion models such as SDXL and MLLMs like LLaVA, it inherently shares the limitations of these foundation models. Hummingbird still faces challenges with complex reasoning tasks such as numerical calculations or code generation due to the symbolic logic limitations inherent to SDXL. Additionally, during inference, the MLLM context descriptor occasionally generates incorrect information or ambiguous descriptions initially, which can lead to lower fidelity in the generated images. We have included qualitative examples to further illustrate these observations, see Figure 7 in the revised version of Appendix B.
>
> Hummingbird currently focuses on single attributes like count, position, and color as part of the multimodal context. This is because this task alone poses significant challenges to existing methods, which Hummingbird effectively addresses. A potential direction for future work is to broaden the applicability of Hummingbird to synthesize images with multiple scene attributes in the multimodal context as part of compositional reasoning tasks.

---

> ### Author Response · Authors · 2024-11-25
> **Response to Reviewer Ag2o (2/2)**
>
> > **Q3 - Would the model maintain robust performance when using alternative, less powerful MLLMs or other multimodal context encoders in place of BLIP-2?**
>
> Thank you for suggesting the comparison. Our design choice to leverage BLIP-2 QFormer in Hummingbird as the multimodal context evaluator facilitates the formulation of our novel Global Semantic and Fine-grained Consistency Rewards. These rewards enable Hummingbird to be effective across all tasks as seen in the table below. When replacing it with a less powerful multimodal context encoder such as CLIP ViT-G/14, we can only implement the global semantic reward as the cosine similarity between the text features and generated image features. As a result, while the setting can maintain performance on coarse-level tasks such as Scene and Existence, there is a noticeable decline on fine-grained tasks like Count and Position. This demonstrates the effectiveness of our design choices in Hummingbird and shows that using less powerful MLLMs, without the ability to provide both global and fine-grained alignment, affects the fidelity of generated images.
>
>
> | **MLLM Name**               | **Method**         | **Existence ACC** | **Existence ACC+** | **Count ACC** | **Count ACC+** | **Position ACC** | **Position ACC+** | **Color ACC** | **Color ACC+** | **Scene ACC** | **Scene ACC+** |
> |-----------------------------|--------------------|-------------------|--------------------|---------------|----------------|------------------|-------------------|---------------|----------------|---------------|----------------|
> | **LLaVA v1.6 7B**           | w/ our Evaluator  | **96.67**         | **93.33**          | **83.33**     | **70.00**      | **81.67**        | **66.67**         | **95.00**     | **93.33**      | **87.75**     | **74.00**      |
> |                             | w/ CLIP           | **96.67**         | **93.33**          | 81.67 (-1.66)        | 66.67 (-3.33)         | 80.00 (-1.67)           | 63.33 (-3.34)             | 95.00         | 90.00 (-3.33)         | **87.75**     | 73.50 (-0.50)         |
> | **InternVL 2.0 8B**         | w/ our Evaluator  | **98.33**         | **96.67**          | **86.67**     | **73.33**      | **78.33**        | **63.33**         | **98.33**     | **96.67**      | **86.25**     | **71.00**      |
> |                             | w/ CLIP           | **98.33**         | **96.67**          | 81.67 (-5.00)        | 70.00 (-3.33)         | 76.67 (-1.66)           | 60.00 (-3.33)            | 96.67 (-1.66)         | 93.33 (-3.34)         | 86.00 (-0.25)        | 71.00          |
>
> ---
>
> > **Q4 - Could the method be adapted for tasks involving more nuanced or abstract text guidance beyond factual scene attributes, such as visual structures (e.g., relative positioning of objects) or style?**
>
> We covered visual structures via the MME Position task for which we conducted experiments and showed results in the main paper, Section 5.3, Table 1. To further explore the method's ability to work on tasks involving more nuanced or abstract text guidance beyond factual scene attributes, we evaluate Hummingbird on an additional task of MME Artwork. This task focuses on image style attributes that are more nuanced/abstract such as the following question-answer pair -- Question: "Does this artwork exist in the form of mosaic?", Answer: "No".
>
>
> Table below summarizes the evaluation. We can observe that Hummingbird outperforms all existing methods on both ACC and ACC+, demonstrating its effectiveness in generating images with high fidelity (in this case, image style preservation) compared to existing methods. This validates that Hummingbird can generalize to tasks involving abstract/nuanced attributes such as image style. We have also included a qualitative comparison for the MME Artwork task in Appendix F, Figure 11.
>
>
> | **Method**             | **Real only** | **RandAugment** | **Image Variation** | **Image Translation** | **Textual Inversion** | **I2T2I SDXL** | **Hummingbird** |
> |-------------------------|---------------|-----------------|---------------------|------------------------|------------------------|----------------|-------------|
> | **Artwork ACC**        | 69.50         | 69.25           | 69.00              | 67.00                 | 66.75                 | 68.00         | **70.25**   |
> | **Artwork ACC+**       | 41.00         | 41.00           | 40.00              | 38.00                 | 37.50                 | 38.00         | **41.50**   |

---

### Author Response · Authors · 2024-11-25
**Response to all reviewers and Thank you!**

We sincerely thank all the reviewers for their thoughtful feedback. We are encouraged that they found our work to be novel (Ag2o, spMH, WWqb) with interesting framework, unique approach (Ag2o), and high originality (spMH). Moreover, the reviewers also acknowledged the significance of our paper: first work and pioneering study on the potential of synthetic data (g5Ub), crucial for many complex visual tasks (spMH, WWqb), excellent performance and impressive results (Ag2o, g5Ub, spMH, WWqb), comprehensive evaluation and detailed analysis (Ag2o), and well-conducted experimental design (spMH).


We address each reviewer's comments individually. We have also incorporated their feedback in the revised manuscript with the following changes (highlighted in blue color):
* Main paper, Subsection 4.2. Fine-Grained Consistency Reward: clarified more details of the ITM classifier and positive class (Ag2o)
* Main paper, Introduction: clarified that Hummingbird is designed as a general-purpose image generator (g5Ub)
* Main paper, Introduction: stated more clearly about experiments on object-centric benchmarks like ImageNet and its OOD variants (spMH, WWqb)
* Appendix B: added Limitations and Future Work (Ag2o)
* Appendix E: added ablation study on the robustness of BLIP-2 QFormer (Ag2o)
* Appendix F: added additional experiments on the Artwork task (Ag2o, spMH, WWqb)
* Appendix G: added additional experiments on the Visual Reasoning task (spMH, WWqb)
* Appendix H: added FID scores to evaluate image quality (g5Ub)
* Appendix I: added User study experiment to evaluate fidelity (g5Ub)
* Appendix J: added training performance on Bongard HOI dataset (g5Ub)
* Appendix K: added the analysis of the number of random seeds (g5Ub)
* Appendix L: added more detailed explanation of Multimodal Context Evaluator (g5Ub, spMH)
* Appendix M: added clarification for the choice of Text Encoders (g5Ub)
* Appendix N: added explanation of how to use Textual Inversion for data augmentation (g5Ub)
* Appendix O: added convergence curves of the training process (g5Ub)

We hope the reviewers will find these modifications helpful, and we are open to more feedback and suggestions.

---

### Author Response · Authors · 2024-12-04
**Thank you for your feedback!**

We sincerely thank all the reviewers for your thoughtful feedback and constructive suggestions for our work. Your insights have been invaluable in improving the quality and clarity of our paper, and we deeply appreciate the time and effort you dedicated to this review process. Thank you!

---

### Meta-Review · Area_Chair_CAzx · 2024-12-18

**Metareview:**

The paper introduces Hummingbird, a diffusion-based image generator that aligns generated images with multimodal inputs (reference image and text) to achieve high fidelity and diversity. It employs a Multimodal Context Evaluator with Global Semantic and Fine-grained Consistency Rewards, validated through new benchmarks and showing superior results over state-of-the-art methods. The paper is recognized by the reviewers that it has a novel framework for balancing fidelity and diversity on MME and HOI datasets, and extensive ablations and user studies. The weakness includes: 1) Limited evaluation of broader datasets and general tasks. 2)Dependency on multimodal context quality.

The paper offers a novel and well-executed solution with strong empirical results and thorough analyses, addressing reviewer concerns effectively.

**Additional Comments On Reviewer Discussion:**

Reviewers raised concerns about reward design clarity, dataset diversity, task generalizability, and computational efficiency. Authors provided mathematical derivations, additional experiments on ImageNet variants, user studies, ablations, and clarified ambiguity handling. These comprehensive responses resolved key concerns, showcasing strong contributions and robustness, leading to a recommendation for acceptance.

---

### Decision · Program_Chairs · 2025-01-22

Accept (Poster)